# B cells sustain inflammation and predict response to immune checkpoint blockade in human melanoma

Johannes Griss [1,2], Wolfgang Bauer [1], Christine Wagner[1], Martin Simon[1], Minyi Chen[1], Katharina Grabmeier-Pfistershammer[1,3], Margarita Maurer-Granofszky [1,12], Florian Roka[1], Thomas Penz[4], Christoph Bock [4,5], Gao Zhang [6,13], Meenhard Herlyn[6], Katharina Glatz[7], Heinz Läubli [8], Kirsten D. Mertz[9], Peter Petzelbauer [1], Thomas Wiesner[1], Markus Hartl [10], Winfried F. Pickl[11], Rajasekharan Somasundaram[6], Peter Steinberger [3] & Stephan N. Wagner [1]

Tumor associated inflammation predicts response to immune checkpoint blockade in human melanoma. Current theories on regulation of inflammation center on anti-tumor T cell responses. Here we show that tumor associated B cells are vital to melanoma associated inflammation. Human B cells express pro- and anti-inflammatory factors and differentiate into plasmablast-like cells when exposed to autologous melanoma secretomes in vitro. This plasmablast-like phenotype can be reconciled in human melanomas where plasmablast-like cells also express T cell-recruiting chemokines CCL3, CCL4, CCL5. Depletion of B cells in melanoma patients by anti-CD20 immunotherapy decreases tumor associated inflammation and CD8$^+$ T cell numbers. Plasmablast-like cells also increase PD-1$^+$ T cell activation through anti-PD-1 blockade in vitro and their frequency in pretherapy melanomas predicts response and survival to immune checkpoint blockade. Tumor associated B cells therefore orchestrate and sustain melanoma inflammation and may represent a predictor for survival and response to immune checkpoint blockade therapy.

[1] Department of Dermatology, Medical University of Vienna, 1090 Vienna, Austria. [2] EMBL-European Bioinformatics Institute, Wellcome Trust Genome Campus, CB10 1SD Hinxton, Cambridge, UK. [3] Institute of Immunology, Center for Pathophysiology, Infectiology and Immunology, Medical University of Vienna, 1090 Vienna, Austria. [4] CeMM Research Center for Molecular Medicine of the Austrian Academy of Sciences, 1090 Vienna, Austria. [5] Department of Laboratory Medicine, Medical University of Vienna, 1090 Vienna, Austria. [6] Molecular & Cellular Oncogenesis Program and Melanoma Research Center, The Wistar Institute, Philadelphia, PA 19104-4265, USA. [7] Institute of Pathology, University Hospital Basel, 4031 Basel, Switzerland. [8] Division of Medical Oncology, University Hospital Basel, 4031 Basel, Switzerland. [9] Institute of Pathology, Cantonal Hospital Baselland, 4410 Liestal, Switzerland. [10] Mass Spectrometry Facility, Max F. Perutz Laboratories (MFPL), University of Vienna, Vienna BioCenter (VBC), 1030 Vienna, Austria. [11] Division of Cellular Immunology and Immunohematology, Institute of Immunology, Center for Pathophysiology, Infectiology and Immunology, Medical University of Vienna, 1090 Vienna, Austria. [12] Present address: Children's Cancer Research Institute, 1090 Vienna, Austria. [13] Present address: Department of Neurosurgery & The Preston Robert Tisch Brain Tumor Center, Duke University Medical Center, Durham, NC 27710, USA. Correspondence and requests for materials should be addressed to J.G. (email: johannes.griss@meduniwien.ac.at) or to S.N.W. (email: stephan.wagner@meduniwien.ac.at)

Cancers such as melanoma, lung, and kidney cancer often present with an inflamed but immunosuppressed tumor microenvironment (TME). Immune checkpoint blocking (ICB) antibodies have significantly improved cancer therapy by overcoming inhibition of T cell effector functions. Yet, a considerable number of patients do not benefit from ICB therapy[1]. It is, therefore, key to understand the mechanisms that regulate inflammation within the TME to develop novel therapies and improve patient survival.

Existing data on the role of tumor-associated B cells (TAB) is inconsistent. Mouse cancer models show that TAB promote tumor inflammation[2,3] but may also inhibit anti-tumor T cell-dependent therapy responses[4–7]. The immuno-inhibitory function of TAB in the latter models resembles that of regulatory B cells (Breg), which are an established source of inhibitory cytokines such as IL-10 and TGF-β (reviewed in ref. [8]). In human cancer, Breg frequencies commonly increase with tumor progression and are enriched in tumors compared to peripheral blood or adjacent normal tissue. Increased IL-10[+] B cell numbers in tumor tissues can also be accompanied by increased numbers of $CD4^+CD25^{+/high}CD127^{low/-}$ and Foxp3$^+$ Tregs[9–12], which are independently associated with tumor progression or reduced patient survival.

In human melanoma, up to 33% of the immune cells can be TAB[13,14]. Phenotypic analyses show the presence of CD20$^+$ TAB (reviewed in ref. [15]) and CD138$^+$ or IgA$^+$CD138$^+$ plasma cells[14,16]. Conclusions about their impact on disease progression and outcome are, however, inconsistent. Similarly, in vivo melanoma transplantation models revealed pro- as well as anti-tumorigenic effects of B cell-deficiency or -depletion in syngeneic mice (reviewed in ref. [17]). In addition, advanced models of human melanoma such as genetically engineered mouse models fail to adequately picture tumor infiltration by TAB[18] and patient-derived xenografts in immune-compromised mice fail to develop a relevant TME. Thus, conclusions about the function of TAB in melanoma can only be drawn from studies in human so far.

In vitro, human melanoma cells provide antigens for prolonged B cell receptor (BCR) stimulation and release inflammation- and B cell-modulating cytokines such as IL-1b and IL-6[19] as well as IL-35[20]. While anti-inflammatory IL-35 promotes the induction of Breg functions[21], pro-inflammatory IL-1b and IL-6 have also been shown to indirectly or directly promote activation and differentiation of B cells[22,23]. We recently showed that human melanoma cells release FGF-2, which induces TAB to secrete IGF-1, a source of acquired drug resistance of melanoma cells to mitogen-activated protein kinase (MAPK) inhibitors[13]. Consistently, clinical data from our pilot trial and an independent case series indicate objective tumor responses and clinical benefit through B cell depletion by anti-CD20 antibodies in end-stage, even MAPK inhibitor-resistant melanoma patients[13,24]. This MAPK inhibitor resistance frequently co-evolves with fundamental TME changes in inflammation and immune cell infiltration and polarization[25].

Here we investigate the impact of TAB on the melanoma TME and its response to ICB therapy (Supplementary Fig. 1). Human melanoma secretomes induce a plasmablast-like phenotype in autologous B cells in vitro with expression of both pro- and anti-inflammatory molecules. Independent single-cell (sc) RNA-sequencing (seq) data confirm this phenotype in human melanoma metastases together with the expression of chemokines critical for immune cell recruitment. Consistently, depletion of TAB from the TME drastically decreases overall inflammation and immune cell numbers. Vice versa, the frequency of plasmablast-like TAB in pre-therapy melanoma samples correlates with improved response and patient survival to ICB therapy.

## Results

**Melanoma TAB are composed of multiple phenotypes.** We used seven-color multiplex immunostaining (CD20, CD19, CD5, CD27, CD38, CD138, and DAPI) to approximate six different TAB subpopulations (Fig. 1a, b) in whole tissue sections of melanoma metastases from 41 different patients (Fig. 1a, Supplementary Data 1). In total, we observed plasmablast-like TAB in 41 samples (100%, Fig. 1c), plasma cell-like TAB in 37 samples (90%, Fig. 1d), activated B cell-like TAB in 37 samples (90%), germinal-center B cell-like TAB in 35 samples (85%), transitional/Breg-like TAB in 27 samples (67%), and memory B cell-like TAB in 22 samples (54%). TAB were primarily located at the invasive tumor-stroma margin in line with previous reports[13,14,26]. This suggests a preferentially contact-independent communication between melanoma cells and TAB.

**Human melanoma secretomes induce Nuclear Factor Kappa B activation in TAB.** The location of TAB at the invasive tumor-stroma margin suggests that melanoma cells communicate with TAB through soluble factors. We therefore exposed immortalized peripheral blood- and tumor-derived B cells derived from four patients with metastatic melanoma to melanoma-conditioned medium (MCM) collected from autologous early passage melanoma cells. As we were unable to establish a tumor cell line for a fifth patient, we additionally exposed this patient's B cells, as well as the B cells of all other patients with MCM from patient 1. From these B cell cultures, we were able to generate RNA-seq ($n = 11$) and complementary proteomic ($n = 8$) profiles.

Seven-hundred and fifty-two genes/proteins were identified as significantly regulated by MCM in both the RNA-seq and proteomics results. Additionally, 802 proteins were found only significantly regulated in the proteomics data and 4231 genes found only significantly regulated in the RNA-seq data (Benjamini–Hochberg (BH) corrected $p$-value $< 0.05$, Supplementary Data 2). There was no marked difference between peripheral blood- and tumor-derived B cells with no significantly differentially expressed proteins observed in the proteomics data (Fig. 2a). The estimated fold changes of genes identified as significantly regulated in both approaches showed a high linear correlation (Spearman cor $= 0.77$, $p < 0.01$, Supplementary Fig. 2A). The subsequent pathway analysis (see Methods) showed upregulation of pathways associated with inflammation, immunity, BCR signaling and intracellular signal transduction, and downregulation of pathways associated with cell cycle, cell division, DNA replication, DNA repair, translation, and transcription (Supplementary Data 2).

One of the most significantly upregulated pathways was tumor necrosis factor (TNF) signaling via Nuclear Factor Kappa B (NF-κB). NF-κB is a key transcription factor for B cell activation in inflammation and immune response[27]. This pathway includes, among others, CD69, CD80, CD30 (TNFRSF8), and CD137 (4-1BB/TNFRSF9), which were significantly upregulated in our proteomics and transcriptomics data (with CD30 (TNFRSF8) among the top-20 upregulated genes) and are known to be upregulated in B cells upon activation. Even though TNF itself showed no significant differences, TNF alpha-induced protein 2 (TNFAIP2) was significantly upregulated as indirect evidence for TNF signaling. In addition, we observed the upregulation of CD40 signaling genes, which also acts via NF-κB. We also found increased phosphorylation of Protein Kinase C Beta (PRKCB), which plays a key role in B cell activation by regulating BCR-induced NF-κB activation (with no significant differences in transcriptomics or global proteomics analysis). Similarly, we observed an increased phosphorylation of NF-κB Activating Protein (NKAP), which is involved in TNF- and IL-1-induced

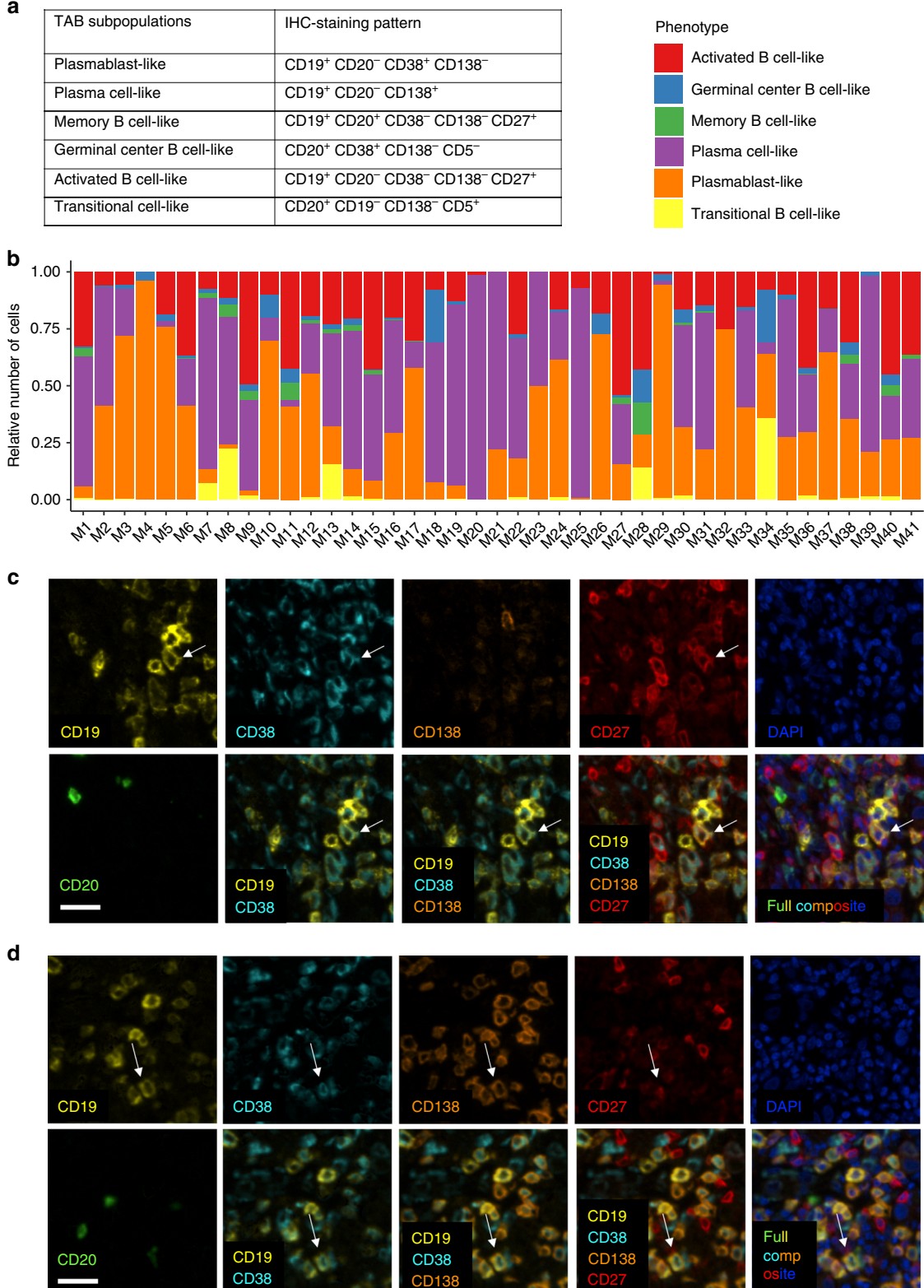

**Fig. 1** Melanoma TAB are enriched for plasmablast- and plasma cell-like phenotypes. **a** Marker combinations used to identify TAB subpopulations by seven-color multiplex immunostaining. **b** Relative frequency of different TAB phenotypes in whole tissue sections of 41 metastatic melanomas **c**, **d** Multiplex immunostaining identifies CD19+CD20−CD38+CD138−CD27+ plasmablast-like (**c**) TAB and CD19+CD20−CD38+CD138+CD27+ plasma cell-like (**d**) TAB. Here, serial images display the same cells from a stromal area at the invasive tumor margin. A full composite image together with DAPI nuclear staining (bottom right) and images for each of the individual markers and different combinations from the composite image are shown. Arrow depicts one of several plasmablast-like (**c**) and plasma cell-like (**d**) TAB. Scale bars represent 20 μm

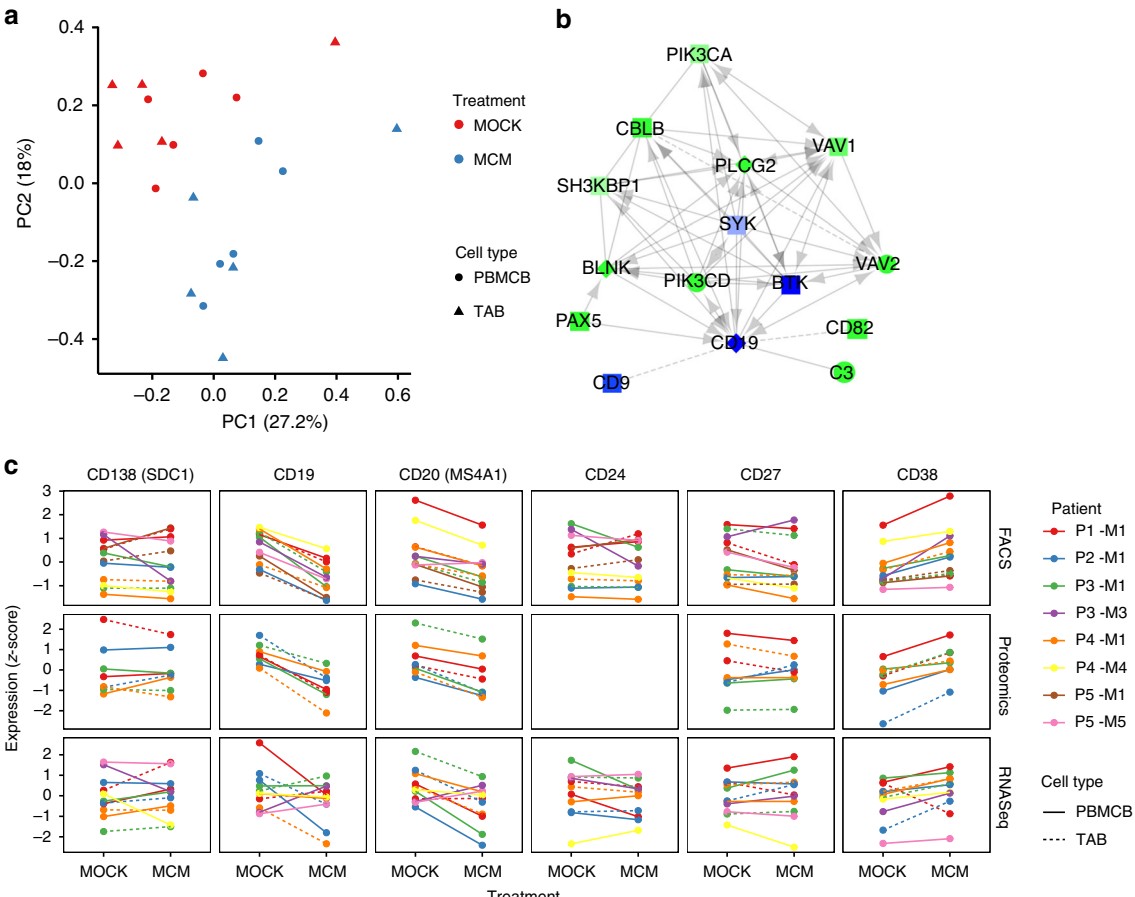

**Fig. 2** Melanoma secretomes induce distinct phenotypic changes in TAB. **a** Principal component analysis of all samples based on the proteomics data comparing samples treated with melanoma-conditioned medium (MCM, blue) against control treated ones (MOCK, red). Peripheral-blood-derived B cells (PBMCB, circles) and TAB (triangles) are shown separately. The percentage in the axis labels represents the variation explained by the component. Points represent one sample per patient and one sample where individual patient samples were pooled together. **b** Interaction network of CD19 with the nearest neighbors depicting the gene abundance estimated by RNA-seq and proteomics. Green represents upregulation, blue downregulation, light colors indicate no relevant difference. Diamonds are genes only detected in proteomics, rectangles genes only detected in RNA-seq, circles genes identified by both methods. For the actual data analysis, the network was constructed using the two nearest neighbors. **c** Expression of key differentiation markers estimated by FACS (geo mean fluoresence-based expression), proteomics, and RNA-seq for melanoma-conditioned medium (MCM) and control (MOCK) treated immortalized peripheral-blood-derived B cells (PBMCB, solid lines) and TAB (dashed lines). All values are shown as z-scores

NF-κB activation. Therefore, melanoma-derived soluble factors directly induce a signaling pattern in B cells associated with activation in inflammation and immune response.

**Melanoma secretomes induce a plasmablast-like rich TAB population.** The downregulation of cell cycle-associated pathways coincided with an unexpected downregulation of CD20 (MS4A1, edgeR $p = 0.05$ RNA-seq, limma $p = 0.08$ proteomics) and CD19 (BH adjusted limma $p < 0.01$ proteomics). This indicates a phenotypic change of B cells. To identify this phenotype, we first used the Reactome functional interaction network[28] and extracted all neighbors of CD19 and their neighbors (Fig. 2b). While known key developmental phenotypes, including reported Breg markers such as CD147, CD24, CD27, CD25, CD39, CD73, or CD138 together with transcription factors BLIMP-1, XBP-1, were not significantly regulated, downregulated (IRF4, CD71), or not detectable (CD5, TIM1), we found a significant upregulation of CD38 (proteomics, BH adjusted limma $p = 0.01$), together with PAX5 (BH adjusted edgeR $p < 0.01$ RNA-seq).

Nine-color FACS staining ($n = 12$) showed changes of CD19, CD20, CD24, CD27, CD38, and CD138 consistent with proteomics and RNA-seq results (Fig. 2c). Immunoglobulin D,

M, and G expression was unchanged (Supplementary Fig. 2B). The induction of CD38 points toward a plasmablast/-cytoid differentiation. This is counter-regulated by increased PAX5 expression. These findings correlate with the high proportion of plasmablast-like TAB in human melanomas (Fig. 1b). Based on these results, we defined a tumor-induced plasmablast-like-enriched B cell population (TIPB) signature with the genes CD27, CD38, and PAX5.

**Melanoma-conditioned TAB express distinct functional signatures.** Cell surface signaling molecules, immuno-stimulatory, and -suppressive cytokines are the most useful factors to distinguish TAB subsets with distinct functions. To profile TIPB on a functional level, we manually extracted six key immunological functional gene sets from the MCM-induced, enriched CD19 interaction network (Fig. 2b, see Methods). To reduce the bias invariably introduced by our small sample size, we additionally extracted highly correlating genes from the The Cancer Genome Atlas (TCGA) skin cutaneous melanoma dataset to create six signatures based on existing literature. We explicitly chose a literature-backed strategy to avoid generation of purely data-driven signatures of unclear and arguable biological significance.

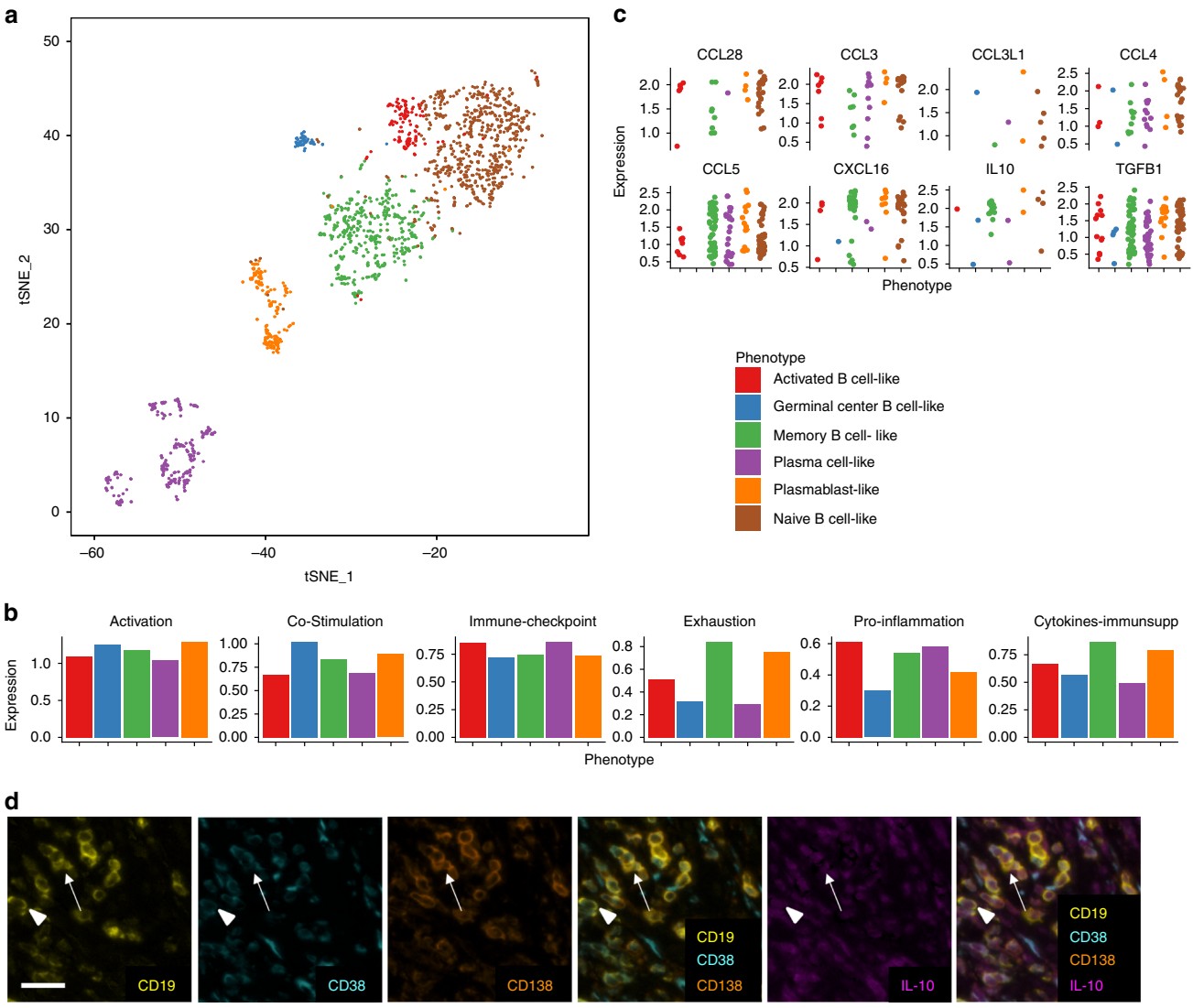

**Fig. 3** Human melanoma scRNA-seq data verify observed functional phenotypes. **a** *T*-distributed stochastic neighbor embedding representation of all B cells with their respective phenotype annotations from the Sade-Feldman et al. dataset[29]. **b** ssGSEA estimated expression of the functional signatures based on the average gene expression per B cell phenotype. **c** Expression of specific chemokines and immunosuppressive cytokines per B cell phenotype and individual cell. **d** Confirmation of IL-10 expression in plasma cell-like (arrow) and plasmablast-like (arrowhead) TAB by quadruple marker immunostaining. Scale bar represents 20 μm

In line with the pathway analysis (increased signaling via NF-κB), these key upregulated gene sets included activation-associated genes (CD69, CD72, CFLAR, FGFR1, SELPLG, CD86), co-stimulatory genes (ICAM3, TNFRSF13C, CD40, CD72, C3, CD80, CD86, CD27, CD28, ICOS, TNFRSF9, CD40LG, ARHG-DIB), and pro-inflammatory genes (TNF, IL12B, IL18, LTA, TNFAIP2, C3, HCK). In addition, immune checkpoint-associated genes (CD274, PDCD1LG2, TNFRSF14/HVEM, LGALS9, BTLA, LAG3, HAVCR2/TIM-3, ADORA2A), immunosuppressive genes (IL10, TGFB1), as well as B cell exhaustion-associated genes (PDCD1, FCRL4, SIGLEC6, CD22) were upregulated.

We used scRNA-seq data by Sade-Feldman et al. on immune-checkpoint blockade and response in human melanoma[29] to validate these functional signatures in vivo. TAB clustered into distinct groups, which we correlated with known developmental B cell phenotypes (Fig. 3a). Overall, all B cell phenotypes expressed comparable levels of our functional signatures (Fig. 3b) but with distinct differences on the single gene level (Supplementary Data 3). Plasmablast-like and memory-like B cells expressed substantial levels of immunosuppressive cytokines such as TGFB1 and IL-10 (Fig. 3c). We could additionally validate the expression of the latter through multiplex immunostaining in plasmablast-like B cells (Fig. 3d). Remarkably, TAB also expressed numerous chemokines including T (reg) cell, and macrophage chemoattractants CCL3, CCL3L1, CCL4, CCL5, CCL28, and CXCL16 (refs. [30–33]) in plasmablast-like B cells (Fig. 3c). Together, these results indicate that TIPB are able to regulate inflammation and shape the cellular composition of the human melanoma TME.

**The TIPB subset correlates with inflammation and survival.** Next, we tested whether the TIPB signature correlates with the identified functional signatures in independent melanoma cohorts and evaluated its effect on clinical outcome in the TCGA skin cutaneous melanoma cohort. All functional signatures (except for the immunosuppressive genes, Spearman correlation 0.49, BH adjusted $p < 0.01$) showed a strong linear correlation with our TIPB signature (Spearman correlation $\geq 0.78$, BH adjusted $p < 0.01$, Supplementary Fig. 3A). Our TIPB signature

was also significantly correlated with the expression of CD8A (Spearman correlation 0.87, $p < 0.01$, Supplementary Fig. 3B) as well as with the abundance of $CD8^+$ T cells (Spearman correlation 0.82, $p < 0.01$, Supplementary Fig. 3C) and macrophages (Spearman correlation 0.71, $p < 0.01$) as estimated by xCell[34]. Established signatures describing the inflammatory TME and possible response to anti-PD-1 therapy (tumor inflammatory score[35], interferon gamma[36], T cell exhaustion[37], and T cell effector[38] signatures) additionally showed a high linear correlation with our TIPB signature (Spearman correlation ≥ 0.85, BH adjusted $p < 0.01$, Supplementary Fig. 3D). Finally, high expression (above median) of our TIPB signature was correlated with longer overall survival in the TCGA cohort[39] (Fig. 4a). These results show, that the TIPB and the predicted functional signatures, are highly associated with T cell abundance and inflammation, and, importantly, are linked with patient outcome.

Anti-PD-1 therapy frequently leads to an increase in B cell numbers, which should enhance our functional signatures. We used the transcriptomics data by Riaz et al. containing (partially matched) 51 pre-anti-PD-1 therapy and 58 on-anti-PD-1 therapy samples[40]. In this independent cohort, all signatures with exception of the immunosuppressive genes (Spearman correlation = 0.6, BH adjusted $p < 0.01$) showed a strong linear correlation with the TIPB signature (Spearman correlation >= 0.77, BH adjusted $p < 0.01$, Supplementary Fig. 4A). Again, the TIPB signature correlated with CD8A expression (Spearman correlation 0.9, BH adjusted $p < 0.01$, Supplementary Fig. 4B), estimated $CD8^+$ T cell (Spearman correlation 0.76, $p < 0.01$, Supplementary Fig. 4C) and macrophages (Spearman correlation 0.68, $p < 0.01$) abundance. We also observed a consistent upregulation of all signatures during anti-PD-1 therapy (one-sided $t$-test BH adjusted $p <= 0.03$, df = 100–107, $t = -2.03$– $-2.77$, immunosuppressive genes BH adjusted $p = 0.06$, df = 96.33, $t = -1.58$, exhaustion BH adjusted $p = 0.09$, df = 106.7, $t = -1.35$, Fig. 4b). Interestingly, high expression of our TIPB signature before therapy (top 25% versus lower 25%) predicted overall survival (Likelihood ratio test $p = 0.06$, Fig. 4c). While the TIPB signature did not correlate with clinical response (Supplementary Fig. 5) in this dataset, scRNA-seq showed plasmablast-like and naive-like B cell frequencies in pre-therapy tumor samples to be significantly higher in patients responding to ICB therapy (two-sided Wilcoxon rank sum test, BH adjusted $p$-value = 0.04, Fig. 4d).

In order to assess the functional impact of TIPB on T cells, we performed a surrogate co-culture assay with MCM-induced B cells and NF-κB-reporter Jurkat T cells in vitro. There, MCM-induced B cells significantly increased the effect (NF-κB promoter activity) of PD-1 blockade on PD-1-expressing Jurkat T cells (two-sided $t$-test, BH adjusted $p$ 0.02–0.06, see Methods, Fig. 4e). Additionally, MCM increased B cell viability (Supplementary Fig. 6). Together, these functional data support the clinical importance of the identified TIPB population.

**Loss of TAB reduces melanoma-associated inflammation**. We evaluated the loss of TAB in a cohort of patients with metastatic melanoma treated with anti-CD20 antibodies[13,41] (see Methods, Supplementary Fig. 1). The dataset consists of nine patients with pre- and on-anti-CD20 therapy samples (therapeutic setting) and two patients with pre- and on-therapy samples, where the metastases developed de-novo in B cell-depleted patients on therapy[41] (adjuvant setting, Supplementary Data 1). Out of these 11 patients, matched pre- and on-therapy samples of six patients could be characterized using whole-tissue RNA-seq. Principal component analysis showed no systematic difference between the two patient groups (Fig. 5a).

Next to the expected downregulation of CD19 and CD20 (MS4A1), all patients showed a consistent, significant downregulation of CD8A on anti-CD20 therapy (BH adjusted edgeR $p = 0.02$, Supplementary Data 4). This was linked to a consistent reduction of $CD8^+$ T cells (one-sided, paired $t$-test, BH adjusted $p = 0.07$, $t = 1.74$, df = 5) and macrophages (one-sided, paired $t$-test, BH adjusted $p = 0.07$, $t = 2.17$, df = 5, Fig. 5b). A signature consisting of CD4 and FOXP3 to approximate the abundance of $FOXP3^+$ T cells also decreased significantly (one-sided, paired $t$-test $p = 0.05$, df = 5, $t = 2.03$, Fig. 5b).

Anti-CD20 therapy caused a significant downregulation of our TIPB signature and all our associated functional signatures (one-sided, paired $t$-test, BH adjusted $p$-value 0.01–0.04, $t = 2.3$–4.8, df = 5, Supplementary Fig. 7A). Published signatures describing melanoma-associated inflammation and possible response to anti-PD-1 therapy (tumor inflammatory score[35], interferon gamma[36], T cell exhaustion[37], and T cell effector[38] signatures) highly correlated with our TIPB signature (Spearman correlation 0.7–0.9, BH adjusted $p < 0.01$, Supplementary Fig. 7B) and showed a significant decrease on anti-CD20 therapy (one-sided, paired $t$-test, BH adjusted $p = 0.01$, df = 5, $t = 3.11$–5.46, Fig. 5c).

We further used six-color multiplex immunostaining to characterize and quantify B and T cells from a total of 25 tumor samples from the patients of the therapeutic cohort (see Methods). This confirmed that B cell depletion led to a marked reduction of both $CD8^+$ and $CD4^+$ T cells (one-sided Wilcoxon rank sum test, BH adjusted $p < 0.01$, 0.06 respectively) at the invasive tumor-stroma margin without significantly affecting total cell numbers (two-sided $t$-test, $p = 0.9$, $t = -0.02$, df = 52, Fig. 6a, Supplementary Fig. 8A, B). Loss of B cells also coincided with a significant reduction in the total area of tertiary lymphoid structures (TLSs) (one-sided paired $t$-test $p = 0.04$, $t = 2.5$, df = 3, Supplementary Fig. 8C, D). This effect was stable over at least 6 months (Fig. 6b, Supplementary Fig. 8E). In one patient where samples were available over a timespan of close to 2 years, a short influx of T cells was observed around week 25, shortly prior to disease progression and termination of anti-CD20 therapy (Fig. 6b). Upon termination of subsequent (poly-)chemotherapy (week 76, due to continued disease progression), $CD19^+$ B cells including plasmablasts, plasma cells, and activated B cells as well as T cells reoccurred. In on- and after-anti-CD20 therapy tumor samples where we were able to detect T cells, we did not observe an effect on tissue-resident, central and effector memory T cell composition (Supplementary Fig. 9).

These data validate the essential role of TIPB to sustain tumor inflammation and recruit $CD8^+$ T cells in human melanoma.

## Discussion

The current focus in clinical tumor immunology centers on T cells, both in terms of treatment concepts and signatures for patient stratification. Existing data on the role of B cells is controversial with the prevailing concept of Breg that inhibit the inflammatory anti-tumor response. Here we show that TAB are essential to sustain the inflammatory TME through melanoma-induced subpopulations enriched for plasmablast-like cells. In our own and independent scRNA-seq data, these subpopulations simultaneously express pro- and anti-inflammatory factors pointing toward a more complex role of TAB.

The co-expression of immunostimulatory and -inhibitory cytokines and cell-surface receptors characterizes TIPB as a rather unique B cell population. Melanoma-induced stimulatory signatures indicated antibody-independent functions of TIPB, such as antigen presentation, T cell activation, and cytokine production with the ability to promote local inflammation as recently shown in multiple sclerosis[42]. TIPB also expressed

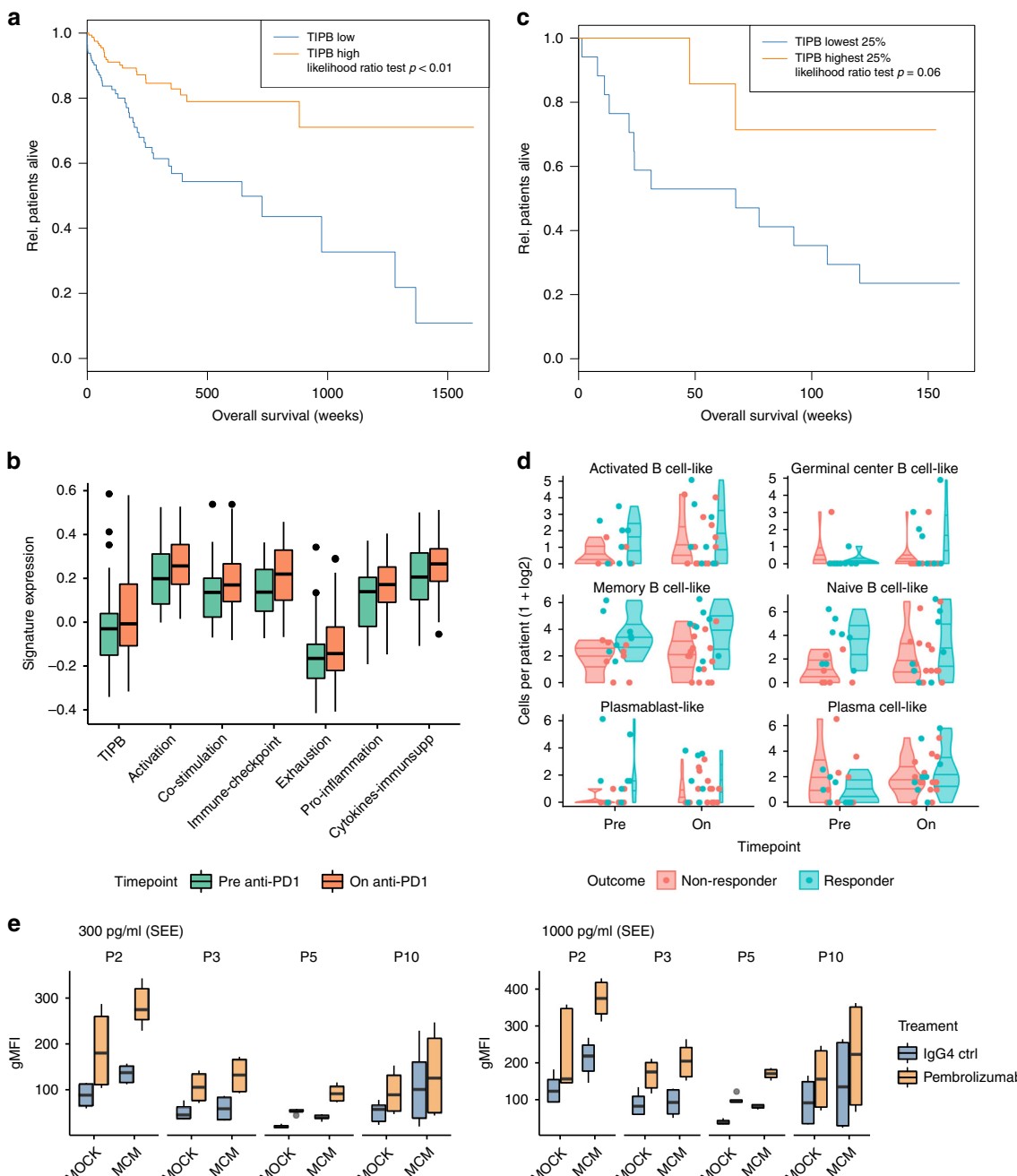

**Fig. 4** TIPB predict improved survival of melanoma patients. **a** Survival analysis of patients expressing high (orange line) or low levels (blue line) of the TIPB signature separated by the median expression in the TCGA melanoma cohort. **b** Expression of the TIPB signature and all functional signatures before (green boxes) and on (orange boxes) anti-PD-1 therapy as estimated in the Riaz et al. dataset[40]. **c** Survival analysis based on the TIPB signature in the pretreatment samples comparing top 25% expressing samples (organe line) against the lower 25% samples (blue line) in the Riaz et al. dataset[40]. **d** Frequencies of B cell phenotypes (logarithmic scale) before (red) and on (blue) ICB therapy in the scRNA-seq dataset from Sade-Feldman et al. separated by response[29]. Plots represent the relative frequencies, lines represent the 25%, 50%, and 75% quantiles. **e** NF-κB activity of PD-1-expressing Jurkat T cells co-cultured with control (MOCK) and MCM-conditioned B cells (MCM). Values are shown for control (blue boxes) and Pembrolizumab (orange boxes) treatment. Activation was measured by geo mean fluorescence intensity (gMFI) after stimulation with 300 pg/ml (left panel) and 1 ng/ml (right panel) *Staphylococcus* Enterotoxin E (SEE). In all boxplots, lower and upper hinges correspond to the first and third quartiles, center line to the median. Upper whisker extends from the hinge to the largest value no further than 1.5 times the interquartile range

inhibitory receptors and ligands as well as IL-10 and TGF-beta as indicated by scRNA-seq and multiplex immunostaining data. In this respect, TIPB resemble Breg, which directly and indirectly suppress proliferation, anti-tumor activity, and differentiation of several immune cells residing in the TME[43–48]. The expression of these inhibitory factors in TIPB is also in line with

previous reports on human CD27^intCD38^+CD138^− plasmablasts as a major IL-10^+ B cell population[45], the suppression of IL-17A^+CD4^+ T cells[49] in human colorectal cancers being highly enriched for IL-10^+CD19^loCD27^hi plasmablasts, and murine IgA^+PD-L1^+IL-10^+ plasmocytes that prevent CD8^+ T cell-mediated regression of hepatocellular carcinoma[7] and

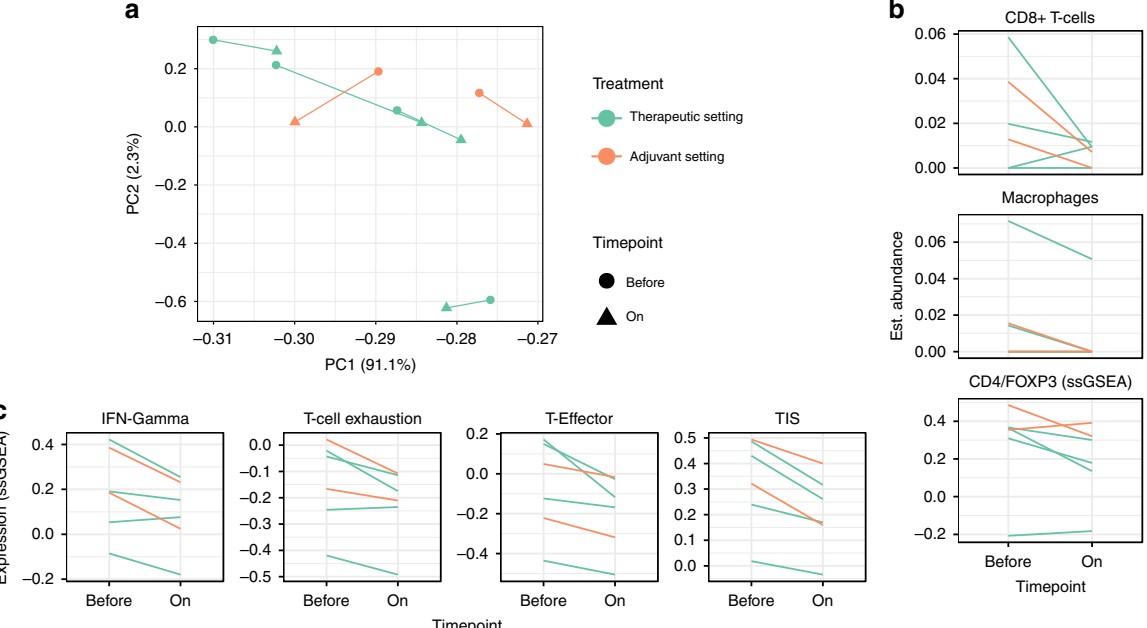

**Fig. 5** Depletion of TIPB reduces tumor inflammation and CD8[+] T cell numbers. **a** Principal component analysis of RNA-seq data from melanoma samples before (circles) and on (triangles) anti-CD20 therapy. On-therapy samples consist of metastases affected by anti-CD20 therapy (therapeutic setting, green lines) and of metastases that developed de novo in B cell-depleted patients (adjuvant setting, orange lines). Percentage numbers in axis labels represent the explained variation by each component. Lines link a patient's samples. **b** xCell estimated abundance of cell types in tissue samples before and on anti-CD20 therapy. Abundance of CD4[+]FOXP3[+] was estimated using ssGSEA since no comparable xCell signature exists. **c** Expression of established inflammation (interferon (IFN) gamma, tumor inflammatory score (TIS), and T cell gene signatures before and on anti-CD20 therapy

impede T cell-dependent chemotherapy of castration-resistant prostate cancer[4]. Our data expand these observations by pin-pointing the additional expression of inhibitory ligands such as LGALS9/Galectin-9, and TNFRSF14/HVEM in TIPB. These ligands can stimulate their cognate inhibitory receptors HAVCR2/TIM-3 and BTLA, which are induced on T cells upon persistent antigen exposure and drive T cell dysfunction.

This seemingly contradictory role of TAB, with both pro- and anti-inflammatory roles is in-line with findings from other diseases. In murine models of autoimmune diseases such as experimental multiple sclerosis and systemic lupus erythematodes[50,51], B cell depletion therapy has opposing effects dependent on the stages of the disease. While B cell depletion in the early phase of disease exacerbates disease severity, it attenuates disease severity in the late phase of the disease. In line with these data, CD20[+] TAB in human oro- and hypopharynx cancer are associated with a favorable outcome in early disease, but a negative outcome in advanced disease[52]. In addition, a favorable prognostic effect of CD20[+] TAB can be linked only to distinct histologic or molecular cancer subtypes, as shown for breast cancer (in ER−, basal, and HER2+, but not triple-negative subtypes[53,54]), ovarian cancer (in high-grade serous, but not other subtypes[55]) and mesothelioma (in the epithelioid, but not non-epithelioid subtype[56]).

Together with our observations, these data emphasize the importance of future context-specific TAB analyses in human cancer considering disease stages, molecular subtypes, and therapies. The tide mode describes expression of co-stimulatory and inhibitory signals as a dynamic system to fine-tune immune responses[57]. Additionally, the immune set point model highlights the often subtle contribution of intrinsic and extrinsic factors that modulate tumor inflammation and tip the balance between immunity and tolerance[58]. In human melanoma, even small variations within the release of tumor antigens/neoantigens, extrinsic or environmental factors, the secretion and

consumption of cytokines and chemokines, distinct genetic sub-types as well as therapeutic agents can tip this balance. The well-known dichotomy of clinical responses to ICB could thus reflect the net outcome of all these drivers and thus the overall inflammatory status of the TME[58]. Similarly, pro-tumorigenic[13] as well as pro-immunogenic functions of TIPB can result from a disbalanced expression of inflammation-modulating factors and depend on the degree of inflammation in the respective TME.

The functional plasticity of TIPB emphasizes the importance of a careful evaluation of their functions in different tumor and host contextes to guide the development of TIPB-targeting strategies for novel cancer combination therapies.

## Methods

**Patient-derived material**. Samples of both trials were collected under local ethics committee-approved protocols (ethical review board of the Medical University of Vienna, votes 457–2007, 969–2010) after obtaining informed patient consent. Separate ethics committee-approval was obtained to make the derived proteomics and RNA-seq data publicly available (ethical review board of the Medical University of Vienna, vote 1555–2016).

Ten patients with metastatic melanoma were treated with the anti-CD20 antibody ofatumumab in a therapeutic setting[13] (Clinical trials number ClinicalTrials.gov NCT01376713). Baseline demographic and clinical characteristics of these patients as well as response data are given in ref. [13], additional clinical data are shown in Supplementary Data 1. Patients had to have adequate biopsy material for (i) multiplex immunohistochemistry (IHC) staining and/or (ii) RNA-seq analysis from time points before initiation of therapy (week 0) and/or under anti-CD20 therapy (week 9 ± 2, with additional later samples as indicated for some patients), when immunophenotyping of peripheral blood mononuclear cells (PBMCs) showed a consistent loss of CD19[+] B lymphocytes (week 2 and later). From nine patients, formalin-fixed paraffin-embedded (FFPE) tumor biopsies could be subjected to multiplex IHC, RNA-seq data were successfully generated from matched samples of four patients. We also collected pre-therapy PBMC samples for generation of EBV-immortalized peripheral B cells[13,59] as well as adequate pre-therapy tumor biopsy material for generation of (i) early passage melanoma cells[59,60] and (ii) autologous EBV-immortalized

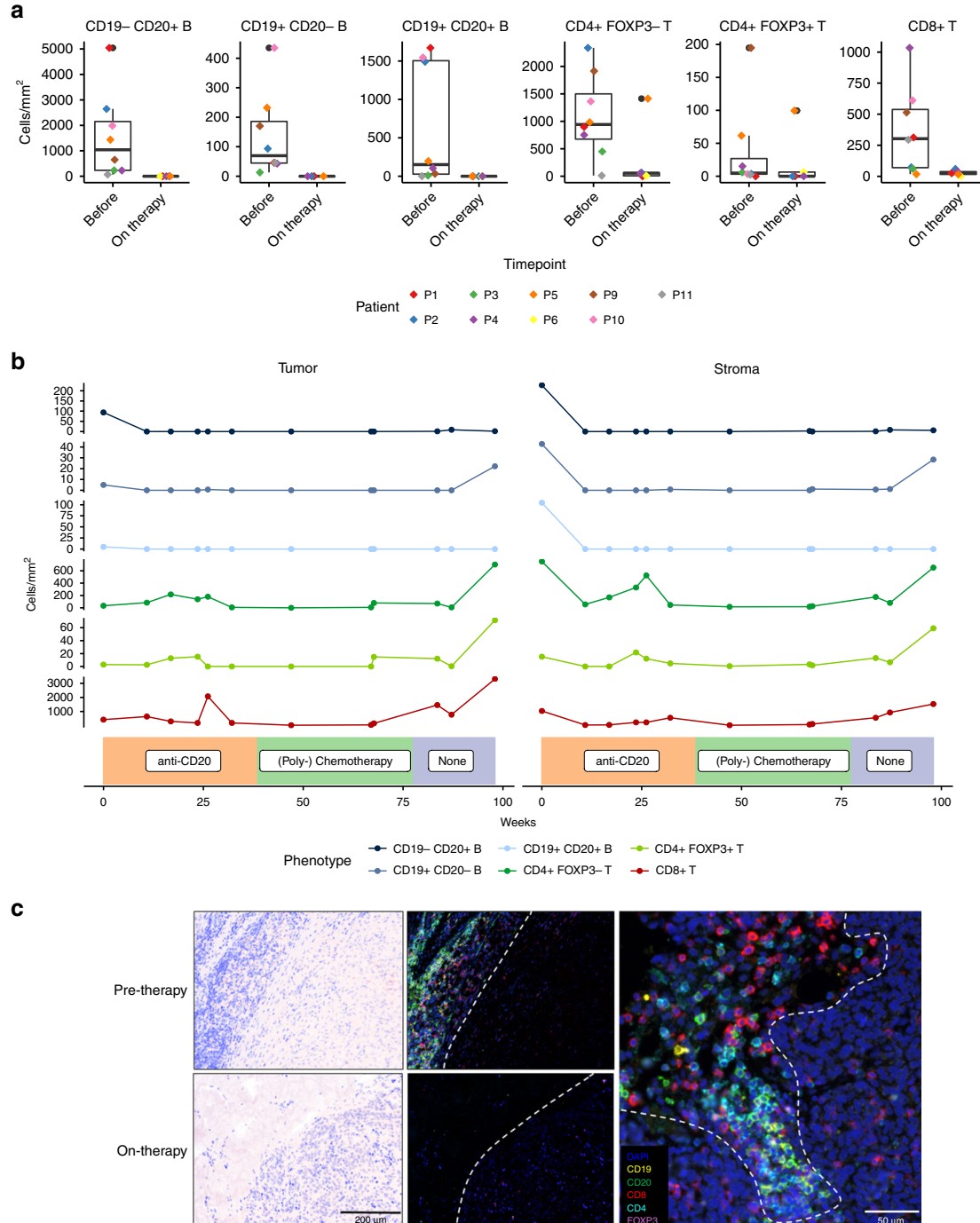

**Fig. 6** Multiplex IHC confirms reduction of CD8+ and CD4+ T cells through B cell depletion. **a** B- and T cell numbers at the invasive tumor-stroma margin quantified using six-color multiplex immunostaining in tumor samples from nine patients obtained before and on therapy (at week 9 ± 2). Lower and upper hinges correspond to the first and third quartiles, center line to the median. Upper whisker extends from the hinge to the largest value no further than 1.5 times the interquartile range. Other data points are shown as outliers (black circles). Individual patient values are shown through differently colored diamonds. **b** Longitudinal analysis of tumor samples obtained over nearly 2 years from patient 4. Sample regions were classified as invasive tumor-stroma margin (Stroma) and intratumoral (Tumor). The lowest panel depicts the respective therapy at the given timepoints. Line colors represent the respective cell types. **c** Multiplex immunostaining for indicated antigens (right) of a representative patient-matched pair of melanoma samples before (top) and on (down) anti-CD20 therapy. Images were taken at the invasive tumor-stroma margin (left: pathology overviews), separated by a dashed line (middle: corresponding immunostaining overviews). Close-up showing a dense B- and T cell infiltrate at the invasive tumor-stroma margin (right). Note loss of B and T cell infiltrates, but not tumor cellularity on therapy (left and middle, down)

TAB[13,59] from tumor single-cell suspensions. From four patients, all these samples were successfully established, from a further patient EBV-immortalized peripheral B cells as well as EBV-immortalized TAB could be established. To increase the number of patients with pre-therapy early passage melanoma cells and autologous EBV-immortalized peripheral B cells and TAB, we included one further patient who was considered for but did not undergo anti-CD20 therapy.

In addition, of two patients with metastatic melanoma who were treated with the anti-CD20 antibody rituximab in an adjuvant setting[41] (EudraCT number:

2007-005125-30), adequate tumor biopsy material for RNA-seq was available at two time points: before initiation of anti-CD20 therapy, when the tumor mass was completely resected, and after disease progression under therapy. Both studies were investigator-initiated clinical pilot trials. Anti-CD20 antibodies (ofatumumab vs. rituximab) were chosen based on availability. Clinical details are summarized in Supplementary Table 2.

**Induction experiments with MCM.** Immediately after surgical resection, melanoma cells and TAB were collected by mechanical and enzymatic dissociation of metastatic melanoma tissue into single cells (tumor dissociation kit, gentleMACS Dissociator, both Miltenyi Biotec GmbH, Bergisch Gladbach, Germany) as described earlier by us[59,60]. Peripheral blood samples were collected in BD Vacutainer CPT cell preparation tubes and PBMC prepared according to the manufacturer's instructions.

Melanoma cells grown from these single-cell suspensions were frozen after passage 2 and stored in liquid nitrogen. Cell purity was determined by flow cytometric staining for melanoma-associated chondroitin sulfate proteoglycan (MCSP) on a FACS Calibur (Becton Dickinson, San Jose, CA). The percentage of MCSP-positive cells was >97%[60]. Sorting of TAB from single-cell suspensions and peripheral B cells from PBMC was performed with Dynabeads CD19 Pan B (Life Technologies, Grand Island, NY) as described[59]. After detachment of beads and antibodies (DETACHaBEAD CD19 Kit, Thermo Fisher Scientific), flow cytometric analyses with a (PE)-conjugated CD19 antibody (Miltenyi Biotec) revealed a purity of collected B cell samples between 89.3% and 98.4%, respectively. Immortalization of TAB and peripheral B cells with EBV was performed on autologous irradiated ($2 \times 30$ Gy) PBMC as feeder cells essentially as described[59,61]. The purity of immortalized B cells was assessed as >96% in all samples by flow cytometry. A semi-quantitative comparative proteome of autologous freshly isolated vs. immortalized TAB has revealed significant differences predominantly in pathways related to cell cycle proliferation, apoptosis, and interferon response[59]. In addition, we have recently shown that immortalized TAB and peripheral blood-derived B cells can nicely recapitulate the induction of pro-tumorigenic/pro-inflammatory factors/cytokines in freshly isolated TAB and peripheral-blood-derived B cells upon exposure to soluble factors from human melanoma cells[13,59] and, thus, can offer a consistent source of quality cells for such experiments.

To produce MCM, early passage (p3) human melanoma cells were grown in complete RPMI 1640 medium (Life Technologies) supplemented with 10% FCS, 100 IU penicillin G, and 100 mg/ml streptomycin at 37 °C in a humidified atmosphere with 5% $CO_2$. At 80% confluence, the entire medium was replaced with fresh medium. MCM was collected 48 h thereafter and sterile filtrated. Control medium was prepared the same way, but without melanoma cells. For induction experiments, $5 \times 10^5$ human B cells were seeded in 24-well cell-culture plates (Corning Costar) and incubated in 1 ml MCM or control medium for 48 h. About 50% of the medium was substituted with fresh MCM after 24 h. B cells were then pelleted by centrifugation at $400 \times g$ for 5 min at RT, snap-frozen and stored at −80 °C.

**FACS analysis.** Mock- or MCM-treated immortalized B cells or TAB were stained with the following antibodies or matched isotypes and analyzed on a FACS Aria III (BD): CD19 BV711 (clone SJ25C1, 0.06 µg/100 µl, catalog number 563036), CD20 AF700 (clone 2H7, 0.5 µg/100 µl, 560631), CD24 PE-CF594 (clone ML5, 1 µg/100 µl, 562405), CD27 BV421 (clone M-T271, 0.25 µg/100 µl, 562513), CD38 APC (clone HIT2, 0.125 µg/100 µl, 555462), CD138 PE (clone MI15, 0.125 µg/100 µl, 552026), IgD PE-Cy7 (clone IA6-2, 0.125 µg/100 µl, 561314), IgG FITC (clone G18-145, 0.125 µg/100 µl, 555786), IgM BV605 (clone G20-127, 0.5 µg/100 µl, 562977) (all BD biosciences). Live/dead cell exclusion was performed by addition of 7-AAD (5 µg/ml, Calbiochem) prior to acquisition of the samples. Data were analyzed using FlowJo 10.4.2 (FlowJo LLC). The gating strategy is shown in Supplementary Fig. 10.

**Jurkat reporter assay.** Jurkat E6.1 NF-kB::eGFP and Jurkat E6.1 NF-kB::eGFP-PD-1 reporter T cells have been previously described in detail[62]. For functional assays, reporter cells ($5 \times 10^4$/well) and mock treated or MCM-treated EBV immortalized B cells ($2 \times 10^4$/well) were co-cultured in the presence of *Staphylococcus* enterotoxin E (SEE, Toxin technology, Inc. Sarasota, FL; used at final concentrations of 300 pg/ml and 1 ng/ml) for 24 h. The PD-1 antibody pembrolizumab (KeytrudaTM, MSD SHARP & DOHME GmbH) or an isotype control antibody (Ultra-LEAF human IgG4 isotype control, Biolegend, San Diego, CA) were added at a final concentration of 10 µg/ml. Following 24 h of co-culture, reporter gene expression was assessed as described in detail[62]. Briefly, cells were harvested and stained using a CD19-APC antibody for exclusion of EBV immortalized B cells. Cells were analyzed using a FACS-CALIBUR flow cytometer and FlowJo software (Franklin Lakes, NJ). Geometric mean of fluorescence intensity (gMFI) of viable reporter cells was used for further analysis.

**Cell viability assay, cell numbers.** Cell viability was assayed along[63]. Briefly, B cells were plated in 96-well plates in triplicates and cultured in control medium or MCM (see above) for 24–96 h. Cell viability was determined by cellular dehydrogenase activity using water soluble WST-8 (2-(2-methoxy-4-nitrophenyl)-3-(4-nitrophenyl)-5-(2,4-disulfophenyl)-2H-tetrazolium), monosodium salt according to manufacturer's instructions (Dojindo EU GmbH, Munich, Germany); 20 µl of WST-8 reagent were added to 200 µl of culture volume per well and incubated for 4 h at 37 °C, 5% $CO_2$. The absorbance was determined using a microplate reader (Bender MedSystems GmbH, Vienna, Austria) at 450 nm with 620 nm as a reference wavelength.

Cell numbers were determined by counting B cells plated in 96-well plates at equal cell numbers in triplicates and cultured in control medium or MCM (see above) for 24–96 h. Cells were counted on a FACSCalibur (BD, San Diego, CA).

**Proteomics analysis.** All reagents were of analytical grade and obtained from SIGMA-Aldrich, unless specified otherwise. Cells were lysed in freshly prepared lysis buffer containing 100 mM Tris/HCL pH 7.6, 2% sodium dodecyl sulfate (SDS), 1 mM sodium vanadate, 1 mM NaF, protease inhibitor (cOmplete$^{Tm}$ EDTA-free), and phosphatase inhibitor (PhosSTOP$^{Tm}$) cocktail tablets (both Roche). Cell extraction and DNA sharing was assisted by sonication and cell debris pelleted by centrifugation at $20,000 \times g$ for 15 min at 20 °C. The supernatant was collected and the total protein concentration was determined using the BCA protein assay kit (Pierce Biotechnology).

Filter-aided sample preparation (FASP) was performed using Amicon Ultra Centrifugal 30-kDa molecular weight cutoff filters (Millipore) essentially according to the procedure described by Wiśniewski et al.[64]. Dithiothreitol (DTT) was added to a final concentration of 0.1 M and the samples heated at 99 °C for 5 min; 200 µl of each protein extract was mixed with 3.8 ml of 8 M urea in 100 mM Tris-HCl, pH 8.5 (UA), in the filter unit and centrifuged at $4000 \times g$ for 30 min at 20 °C to remove SDS. Any remaining SDS was exchanged by urea in a second washing step with 4 ml of UA. Free thiols were alkylated with 2 ml of 50 mM iodoacetamide for 30 min at RT. Afterward, three washes with 3 ml of UA solution and then three washes with 3 ml of 50 mM triethylammonium bicarbonate (TEAB) were performed. Proteins were digested on filters with trypsin (1:50; Trypsin Gold, Promega) in 50 mM TEAB overnight at 37 °C. Digested peptides were collected by centrifugation, acidified with trifluoroacetic acid (TFA), and desalted using Sep-Pak C18 SPE cartridges (50 mg, Waters Corporation) using 80% acetonitrile containing 0.1% TFA for the elution and evaporating the solvent in a vacuum centrifuge.

Isobaric labeling was performed using 10plex tandem mass tag (TMT) reagents (Thermo Fisher Scientific); 200 µg peptide digest per cell line was resuspended in 100 mM TEAB buffer and labeled with 0.8 mg of TMT 10-plex™ reagents (Thermo Fisher Scientific) according to the manufacturer's protocol. After 1 h incubation at room temperature, samples were quenched for 15 min with 8 µl 5% hydroxylamine at room temperature. Labeling efficiency was determined running aliquots of the samples on 1 h LC-MS/MS gradients and standard database searches with TMT-tags configured as variable modifications. Corresponding TMT-labeled samples were pooled, acidified with TFA to a final concentration of 1% TFA and concentrated via Sep-Pak C18 SPE cartridges (200 mg bed volume).

Off-line high-pH reversed phase fractionations was performed essentially according to ref. [65] with a few modifications. Peptides were separated on a Waters Xbridge BEH130 C18 3.5-µm 4.6 × 250 mm column on an Ultimate 3000 HPLC (Dionex, Thermo Fisher Scientific) operating at 0.8 ml/min. Buffer A consisted of $H_2O$ and buffer B consisted of 90% acetonitrile (MeCN), both adjusted to pH 10 with ammonium hydroxide. The gradient was set as follows: 1% B to 30% B in 50 min, 45% B in 10 min, and 70% B in 5 min. Fractions were collected at minute intervals and reduced using a vacuum concentrator. Dried peptides were reconstituted in 2% MeCN, 0.1% TFA, and pooled to a total number of 12 fractions using a concatenation strategy covering the whole gradient[65]. An aliquot of each fraction (5–10 µg) was used for analysis of the global proteome and residual samples were dried in a vacuum concentrator prior to TiO$_2$ phosphopeptide enrichment.

Phosphopeptide enrichment was performed using a modified TiO$_2$ batch protocol. In short, titanium dioxide beads (5 µm; GL Sciences, Japan) were sequentially washed with 120 µl of 50% methanol, 300 µl of ddH$_2$O, and $2 \times 300$ µl binding solvent (1 M glycolic acid, 70% MeCN, 3% TFA). In between, beads were spun down and the supernatant was discarded.

Dried peptides of each of the 12 fractions were individually resuspended in 150 µl binding solvent and incubated with the titanium dioxide beads at a bead to peptide ratio of 1:4 for 30 min at RT under continuous rotation. Bead-bound peptides were washed twice with binding solvent, 2× with washing solvent A (70% MeCN, 3% TFA) and 2× with washing solvent B (1% MeCN, 0.1% TFA). Phosphopeptides were eluted from the beads with $2 \times 150$ µl 0.3 M NH$_4$OH. The eluates were acidified by addition of TFA to a final concentration of 2% and desalted using C18 StageTips[66].

Global proteome and the phosphopeptide fractions were separated on an Ultimate 3000 RSLC nano-flow chromatography system using a pre-column for sample loading (PepMapAcclaim C18, 2 cm × 0.1 mm, 5 µm,) and a C18 analytical column (PepMapAcclaim C18, 50 cm × 0.75 mm, 2 µm, all Dionex, Thermo Fisher Scientific), applying a linear gradient over for 2 h from 2 to 35% solvent B (80% acetonitrile, 0.1% formic acid; solvent A 0.1% formic acid) at a flow rate of 230 nl/min. Eluting peptides were analyzed on an Orbitrap Fusion Lumos mass spectrometer equipped with EASY-Spray™ source (all Thermo Fisher Scientific), operated in a data-dependent acquisition mode with a cycle time of 3 s. FTMS1 spectra were recorded at a resolution of 120k, with an automated gain

control (AGC) target of 200,000, and a max injection time of 50 ms. Precursors were filtered according to charge state (included charge states 2–6 $z$), and monoistopic peak assignment. Selected precursors were excluded from repeated fragmentation using a dynamic window (40 s, ±10 ppm). The MS2 precursor were isolated with a quadrupole mass filter width of 1.2 $m/z$. For FTMS2, the Orbitrap was operated at 50k resolution, with an AGC target of 100,000 and a maximal injection time of 150 ms for global proteome samples and 250 ms for phosphopeptide samples. Precursors were fragmented by high-energy collision dissociation (HCD) at a normalized collision energy (NCE) of 42%.

**Transcriptomics analysis**. The amount of total RNA was quantified using the Qubit Fluorometric Quantitation system (Life Technologies) and the RNA integrity number (RIN) was determined using the Experion Automated Electrophoresis System (Bio-Rad).

For the co-culture experiments, the isolated RNA (1 µg) was processed using the SENSE mRNA-Seq Library Prep Kit V2 (Lexogen; #SKU 001.96) according to the manufacturer's protocol. The libraries were sequenced on an Illumina HiSeq 2500 platform with 50 bp single-end reads to obtain on average 25 million reads per sample.

For the samples of the anti-CD20 study, RNA-seq libraries were prepared with the TruSeq Stranded mRNA LT sample preparation kit (Illumina) using both Sciclone and Zephyr liquid handling robotics (PerkinElmer). Library concentrations were quantified with the Qubit Fluorometric Quantitation system (Life Technologies) and the size distribution was assessed using the Experion Automated Electrophoresis System (Bio-Rad). For sequencing, samples were diluted and pooled into NGS libraries in equimolar amounts. Expression profiling libraries were sequenced on Illumina HiSeq 3000/4000 instruments in 50-base-pair-single-end mode.

Base calls provided by the Illumina Real-Time Analysis (RTA) software were subsequently converted into BAM format (Illumina2bam) before de-multiplexing (BamIndexDecoder) into individual, sample-specific BAM files via Illumina2bam tools (1.17.3, https://github.com/wtsi-npg/illumina2bam).

**Data analysis and statistical information**. All statistical tests were performed using R version 3.5.1 (ref. [67]). The complete R scripts, including the processed input data for all datasets and plots shown in this manuscript, are available as Jupyter notebooks in Supplementary Software 1.

For proteomics data analysis, RAW files were converted to the MGF files using ProteoWizard's msConvert tool (version 3.0.9393) using vendor libraries[68]. Peak list files were searched using MSGF+ (ref. [69]) (version 10089) and X!Tandem[70] (version 2017.2.1.2). The precursor tolerance was set to 10 ppm, fragment tolerance to 0.01 Da for X!Tandem, and machine type to QExactive with HCD fragmentation in MSGF+, and 1 missed cleavage was allowed. TMT tags and carbamidomethylation of C were set as fixed modifications, oxidation of M and deamidation of N and Q, and phosphorylation of S, T, and Y (TiO$_2$-enriched samples only) as variable modifications. Searches were performed against the human SwissProt database (version 17–02), combined with sequences of common contaminants and reversed decoy sequences. Search results were filtered based on the target-decoy strategy at 0.01 FDR if both search engines identified the same sequences, diverging results were discarded, and spectra only identified by one search engines filtered at 0.001 FDR (search engine specific).

For quantitation all spectra with >30% interference based on total ion current were discarded. Isotope impurity correction and normalization of intensity values was performed using the R Bioconductor package isobar[71] (version 1.26). Protein abundance was estimated using the R Bioconductor package MSnbase[72] (version 2.6.3) using the iPQF method and only peptides uniquely mapping to a single protein. Differential expression analysis was performed using the R Bioconductor package limma[73] (version 3.35.3). The linear model included the MS run, patient, cell type (PBMCB vs. TAB) as factors next to the treatment group.

For transcriptomics data analysis, Bam files were converted to FASTQ format using the samtools package (https://github.com/samtools/samtools, version 0.1.19). The first nine amino acids were removed using the fastx_trimmer tool (http://hannonlab.cshl.edu/fastx_toolkit, 0.0.14). Alignment was performed using the STAR aligner[74] (version 2.5.3a, 23104886) with the output set to read counts. Alignment was performed against the Ensembl human genome 38 (version 88). Differential expression was assessed using the Bioconductor R package edgeR[75] (version 3.22). Genes with less than 100 transcripts found in all samples were discarded prior to analysis. The linear model included the patient and cell type next to the treatment.

Pathway analysis was performed using the Camera algorithm[76] as implemented in limma against the Molecular Signatures Database[77] (MSIGDB, v6.1) hallmark[78] and C2 canonical pathways collection. The analysis was performed separately for proteomics and transcriptomics data and merged using the Cytoscape[79] Enrichment Map plugin[80] (version 3.1.0).

Cell type abundance in whole tissue samples was estimated using xCell[34] (R package version 1.1.0). Abundance of custom signatures was estimated using the ssGSEA approach[34,81] as implemented in the Bioconductor R package GSVA[82] (version 1.28.0).

**Public datasets**. To create our curated, key functional signatures, we extracted highly correlating genes from the datasets from cutaneous melanoma samples of TCGA[39] using the Bioconductor R Package geneRecommender (version 1.52). TCGA mRNA expression data were extracted using cBioPortal's R interface using the CGDS-R package[83] (version 1.2.5). Survival analysis was performed using the survival package (version 2.42-6, https://github.com/therneau/survival).

We used the data from Riaz et al. to evaluate the effects of anti-PD-1 therapy on our signatures[40]. The TPM normalized expression values were downloaded from GEO (GSE91061). Signature expression levels were estimated using the ssGSEA approach and cell type abundances estimated using xCell (see above).

scRNA-seq from Sade-Feldman et al.[29] was directly downloaded from GEO as TPM normalized expression values (GSE120575). The complete data analysis was performed in R using Seurat 2.3.4 (ref. [84]). The data were normalized using the "LogNormalize" function with a scale factor of 10,000. Variable genes were detected using the "LogVMR" dispersion function and the "ExpMean" function to calculate means. The x.low.cutoff was set to 0.0125, x.high.cutoff to 3, and the y. cutoff to 0.5. Next, the data were scaled regressing out the patient identifier. Out of 100 calculated principal components, the first 35 were used for further analysis. Cells were clustered using Seurath's graph-based clustering algorithm based on the principal components using a resolution of 0.6. Subsequent cell types were annotated based on canonical markers and all B cell-like cells retained. B cells were again clustered using a resolution of 0.9. Clusters were annotated on the found markers (function FindMarkers, min.pct set to 0.25), as well as the expression of canonical B cell markers. Signature expression levels were estimated using the ssGSEA function (see above) based on the average gene expressions. The complete workflow can be found in Supplementary File 1 as a Jupyter notebook.

**Histology**. Whole tissue sections were obtained from human melanoma metastases almost exclusively derived from cutaneous and subcutaneous sites (Supplementary Table 1 for details). All tumor samples were obtained with informed patients' consent and retrieved from the pathology files of the Medical University of Vienna (ethics vote: 405/2006). Diagnosis was based on the review of hematoxylin and eosin-stained sections from FFPE blocks by two authors of this study (P.P., S.N.W.), respectively. In selected tumor samples, S-100, HMB45, or triple CSPG/β3 integrin/HMB45 melanoma marker immunostainings were done to identify tumor cells and to confirm the diagnosis..

**Multiplex IHC staining**. Four-micrometer sections from full FFPE blocks from metastatic human melanoma and control tonsil tissue were utilized for both, the initial establishment of staining conditions for each individual primary antibody (Ab) and the successive optimization of multiplex staining. In a first step, primary Abs against the following antigens were established as single stains initially on human tonsil tissue and thereafter on metastatic human melanoma tumors.

The used antibodies were CD19 (Rabbit monoclonal, clone EPR5906, 1:500, Abcam, catalog number ab134114), CD20 (Mouse monoclonal, clone L26, 1:2000, Dako, M0755), CD27 (Rabbit monoclonal, clone EPR8569, 1:500, Abcam, ab131254), CD38 (Mouse monoclonal, clone AT13/5, 1:1350, Dako, F7101), CD138 (mouse monoclonal, clone MI15, 1:1350, Agilent, M7228), CD5 (Mouse monoclonal, clone 4C7, 1:600, Novocastra, NCL-L-CD5-4C7), IL-10 (Rabbit polyclonal IgG, 1:800, Proteintech, 20850-1-AP), CD8 (Mouse monoclonal, clone C8/144B, 1:900, Biocare Medical, ACI 3160C), FoxP3 (Rabbit monoclonal, clone D2W8E, 1:400, Cell Signaling Technology, 98377S), CD4 (Mouse monoclonal, clone 4B12, 1:500, Dako, M7310), CD69 (Rabbit polyclonal, HPA050525, 1:100, Sigma-Aldrich, HPA050525), CD103 (Rabbit monoclonal, clone EPR4166(2), 1:4000, Abcam, ab129202), CD45RO (Mouse monoclonal, clone UCHL1, 1:2500, Dako, M0742), CXCL13 (Rabbit polyclonal IgG, 1:1800, Proteintech, 10927-1-AP), CD21 (Rabbit polyclonal IgG, 1:3600, Proteintech, 24374-1-AP), CD23 (Rabbit monoclonal, clone SP23, 1:900, Novus Biologicals, NB120-16702), Bcl6 (Mouse monoclonal, clone 1E6B1, 1:9000, Proteintech, 66340-1-Ig).

In a second step, multiplex immunostains were established essentially as described[85,86]. Random integration of sequential Abs within a multiplex panel may lead to imbalanced signals, incomplete staining through interference with previously applied tyramide signal amplification (TSA), disruption of epitopes, and removal of TSA fluorophores because of repetitive antigen-retrievals at high temperature[85,86]. Therefore, each Ab was tested individually for its optimal position in the sequence of multiplex staining to minimize interference with previous Ab-TSA complexes or by alteration of epitopes.

For multiplex immunostains, 4-µm sections were deparaffinized and antigen retrieval was performed in heated citrate buffer (pH 6.0) and/or Tris-EDTA buffer (pH 9) for 30 min. Thereafter, sections were fixed with 7.5% neutralized formaldehyde (SAV Liquid Production GmbH). Each section was subjected to six successive rounds of Ab staining, each consisting of protein blocking with 20% normal goat serum (Dako) in PBS, incubation with primary Abs, biotinylated anti-mouse/rabbit secondary antibodies and Streptavidin-HRP (Dako, 50003), followed by TSA visualization with fluorophores Opal 520, Opal 540, Opal 570, Opal 620, Opal 650, and Opal 690 (PerkinElmer) diluted in 1X Plus Amplification Diluent (PerkinElmer), Ab-TSA complex-stripping in heated citrate buffer (pH 6.0) and/or Tris-EDTA buffer (pH 9) for 30 min and fixation with 7.5% neutralized formaldehyde. Thereafter, nuclei were counterstained with DAPI (PerkinElmer)

and sections mounted with PermaFluor fluorescence mounting medium (Thermo Fisher Scientific).

Respective stainings without primary antibodies were used as a negative control. At equal Ab-concentrations, TSA-based visualization is expected to yield a higher number of positive cells as compared to conventional immunofluorescence. We therefore established TSA-based visualization of primary Abs on control tonsil tissue, the golden standard for lymphocyte antigen detection in pathology, and performed a comparison for each Ab to validated staining patterns in human tonsil (as to the Human Protein Atlas[87], available from www.proteinatlas.org). Thereafter, we balanced the signal through dilution of the primary Abs to obtain staining levels and cell frequencies comparable to conventional immunofluorescence staining. The dilution of the CD19 antibody was optimized to allowing for the detection of CD19[low] plasma cell-like cells as compared to patterns and frequencies obtained by CD138[+] pooled IgA/IgG[+] stainings on human tonsil and melanoma[14,16]. In multiplex stainings, single primary Ab stainings were run in parallel to control for false-positive results through incomplete Ab-TSA complex-stripping and false-negative results through antigen masking (by incubation with multiple primary Abs, umbrella-effect). Spillover effects were controlled for anti-CD20-Ab stainings on tonsil with different Opal fluorophores by signal detection in adjacent components/channels and thereafter for exposure time settings upon acquisition of multiplex-stained tissue sections.

**Tissue imaging, spectral unmixing, and phenotyping**. Multiplexed slides were scanned on a Vectra Multispectral Imaging System version 2 (Perkin Elmer) as described[85,86]. Briefly, a spectral library from spectral peaks emitted by each fluorophore from single stained slides was created with the Advanced Image Analysis software (InForm 2.4, PerkinElmer) and used for spectral unmixing of multispectral images allowing for identification of all markers of interest. Auto-fluorescence was determined on an unstained representative study sample.

TAB subset phenotyping in whole tumor sections: cells were phenotyped as (i) CD19[+] CD20[−] CD38[+] CD138[−] plasmablast-like, CD19[+] CD20[−] CD138[+] plasma cell-like, CD19[+] CD20[+] CD38[−] CD138[−] CD27[+] memory B cell-like, CD20[+] CD38[+] CD138[−] CD5[−] germinal center B cell like, CD19[+] CD20[−] CD38[−] CD138[−] CD27[+] activated B cell-like, CD20[+] CD19[−] CD138[−] CD5[+] transitional/regulatory cell-like TAB and (ii) other cells. The staining protocol has been optimized for detection of CD19. Though CD19[low] plasma cell-like cells could be detected at significant numbers, they may still be underrepresented in our staining data. The same may also be true for the detection of activated B cell-like cells, as expression of CD27 has been reported to be downregulated on TAB[88,89]. After adaptive cell segmentation, cut-off values for each fluorophore/antibody staining were defined. All phenotyping and subsequent quantifications were performed blinded to the sample identity.

TAB and T cell subset distribution in intra- and extratumoral stromal tumor areas: was performed in whole tumor sections assisted by the automated tissue segmentation tool of the InForm software (v.2.4, PerkinElmer), which had been trained to recognize intratumor and extratumoral stromal areas based on tissue morphology including the presence and absence, respectively, of melanoma cells. Total cell counts were determined by DAPI[+] cell numbers referred to the total tumor area, six-color multiplex immunostaining (CD20, CD19, CD8, CD4, FoxP3, DAPI) was used for the detection of CD20[+]CD19[+], CD20[+]CD19[−], and CD20[−]CD19[+] TAB and CD8[+], CD4[+]FoxP3[−], and CD4[+]FoxP3[+] cells. Further T cell subsets were identified by an additional seven-color multiplex immunostaining and phenotyped according to refs. [90,91] as CD8[+]CD4[−]CD69[+]CD103[+] and CD8[−]CD4[+]CD69[+]CD103[−] cells corresponding to a tissue-resident memory T cell phenotype (T$_{RM}$); CD8[+]CD4[−]CD45RO[+]CD27[+] and CD8[−]CD4[+]CD45RO[+]CD27[+] cells corresponding to a central memory T cell phenotype (T$_{CM}$); CD8[+]CD4[−]CD45RO[+]CD27[−] and CD8[−]CD4[+]CD45RO[+]CD27[−] cells corresponding to an effector memory T cell phenotype (T$_{EM}$); and other CD8[+]CD4[−] or CD8[−]CD4[+] T cells.

TLSs were identified according to Silina et al.[92] by a further seven-color multiplex immunostaining. Primary follicle-like TLS were identified as CD4[+] and CD20[+] lymphocyte aggregates with CXCL13 and CD21 but not CD23 expression; immature secondary follicle-like TLS by additional CD23 expression; and mature secondary follicle-like TLS by accessory expression of Bcl6. In on-anti-CD20 therapy samples, TLS were identified along the same criteria, without the presence of CD20 lymphocyte aggregates. Trainable tissue segmentation was performed using the automated tissue segmentation tool of the InForm software (v.2.4, PerkinElmer). TLS areas (as determined by pixels) were read in whole tumor tissue sections and referred to the total tumor area (as determined by pixels).

**Reporting Summary**. Further information on research design is available in the Nature Research Reporting Summary linked to this article.

## Data availability

RNA-seq data were deposited in the ArrayExpress[93] database at EMBL-EBI (www.ebi.ac.uk/arrayexpress) under accession numbers E-MTAB-7472 (induction experiment) and E-MTAB-7473 (anti-CD20 study). The mass spectrometry proteomics data were deposited to the ProteomeXchange Consortium via the PRIDE partner repository[94] with the dataset identifier PXD011799 (induction experiment).

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

## Acknowledgements

This work was supported by the FWF-Austrian Science Fund (project P31127-B28 to S.N.W. and SFB F4609 to W.F.P.) and received funding from the European Union's Horizon 2020 research and innovation programme under grant agreement No. 788042. We thank Bjoern Wendik (Perkin Elmer) for support in tissue imaging and data acquisition of multiplex immunostainings, Barbara Sternitzky and Claudia Kokesch for technical support in processing human tumor samples, the Biomedical Sequencing Facility at CeMM for assistance with next-generation sequencing, the MFPL Mass Spectrometry team for technical support with proteomics, and the VBCF for providing the mass spectrometry instrument pool. This work was supported in part by NIH grants P01 CA114046, P50 CA174523, U54 CA224070, DoD - PRCRP WX1XWH-16-1-0119 [CA150619], and the Dr. Miriam and Sheldon G. Adelson Medical Research Foundation to M.He.

## Author contributions

Hypothesis and study protocol: J.G., G.Z., M.He., R.S., S.N.W. Clinical study and patient materials: J.G., M.M.G., F.R., P.P., W.F.P., S.N.W. Histopathology: C.W., M.C., K.G., H.L., K.D.M., P.P., S.N.W. FACS and functional assays: W.B., M.S., P.S., K.G.M., S.N.W. Proteomics: J.G., M.M.G., M.Ha., C.W. RNA-seq: J.G., T.W., C.W., T.P., C.B. Bioinformatics: J.G. Manuscript: J.G., S.N.W. All authors contributed to the final version of the manuscript.

## Additional information

**Competing interests:** The authors declare no competing interests.

