## [Peer Review File · Nature Communications]

Reviewers' comments:

Reviewer #1 (Remarks to the Author): Expert in Cancer Immunology

Griss and co-authors use sequencing and immunohistochemistry approaches on new and publically available human datasets to identify a tumor-associated plasmablast-like B cell signature (TIPB) in metastatic melanoma that correlates with patient survival and response to immune checkpoint blockade. Although these data present an interesting potential biomarker for therapy, more mechanistic insight is needed for how tumor associated B cells functionally regulate T cells in the melanoma tumor microenvironment, and how this interaction ultimately may impact clinical response.

1. Is the decrease in TIPB signature on anti-CD20 treatment (Figure 5) associated with decreased overall survival, as may be predicted from data in Figure 4A? If so, how might this be explained, given previous studies (cited in Introduction) demonstrating enhanced anti-tumor response and clinical benefit of anti-CD20 treatment in therapy-resistant metastatic melanoma?
2. Does frequency of intratumoral T cells or T cell subset (resident, central or effector) change on anti-CD20 therapy? While the reduction in CD8+ T cells in extratumoral stroma is striking in anti-CD20 treatment image (Figure 5D), it is less apparent whether intratumoral T cells are significantly decreased. In addition, cellular density appears to be substantially decreased with anti-CD20 treatment – is this representative of all on-treatment tissues evaluated? Quantification is required.

Reviewer #2 (Remarks to the Author): Expert in melanoma and immunology

In their study Griss et al. address the role of tumor-associated B-cells (TABs) for the immune contexture of human melanomas. Using a multiplex immunostaining approach with seven established markers the authors characterize TABs in a cohort of human metastatic melanomas. In particular, they describe and focus on a predominant population of CD19+CD20-CD38+CD138-CD27+ plasmablast-like tumour-associated B-cells (TAB). Because TABs were located at the tumor margin, the authors assumed an indirect communication between melanomas cells and TABs. Exposure of autologous B-cells from peripheral blood to melanoma cell-conditioned medium induced a plasmablast-like B-cell phenotype in a NF- κ B-dependent manner. The authors therefore suggested that melanoma cells educate B-cells in the tumor microenvironment (TME). Showing that these plasmablast-like TABs produce T-cell recruiting chemokines the authors asked whether TABs, in particular plasmablast-like TABs, are involved in T-cell recruitment. Analysis of pre- and on-treatment specimens from melanoma patients treated with an anti-CD20 antibody showed reduced infiltration of CD8+ T-cells and macrophages under anti-CD20 therapy. The work by Griss et al. emphasizes the importance of B-cells for the immune cell composition in melanomas with potential implications for immune checkpoint inhibition as supported by correlative analyses of published gene expression data sets. Indeed, the role of B-cells is largely overlooked in the context of ICI. The authors build on their previous study an expertise in this field. The topic of the present work is timely and of interest to a large scientific community. However, there are several issues that need to be addressed.

#1 The work focuses on a predominant population of plasmablast-like TABs, which are CD20- (or at least CD20low). Key to the study is the analysis of longitudinal biopsies from melanoma patients treated with anti-CD20, which would then not target the plasmablast-like TABs directly or insufficiently. What is the concept of their reduction in on-treatment specimens? Reduced TME-dependent conversion from the general B-cell population? It is important to provide more patient details related to the study and additional mechanistic insights to TIPBs.

-How was the reduction of B-cells in the peripheral blood from patients receiving anti-CD20

therapy at the time when biopsies were taken?

-What is the time frame between the biopsies? What about the clinical response? Did patients progress under therapy? Could the reduction of CD8+ T-cells in the tumor microenvironment be related to disease progression rather than a causal relationship to anti-CD20 therapy and B-cell depletion?

-What about biological changes of plasmablast-like TABs (TIPBs)? Viability, persistence in the TME? It would be important to characterize biological changes (viability etc...) of B-cells exposed to melanoma-conditioned medium as a surrogate of TIPBs.

-What about T-cell regulatory properties of TIPBs? Would B-cells exposed to melanoma-conditioned medium exert T-cell suppressive or T-cell stimulating properties in co-culture assays (aCD3/aCD28 stimulations)? This would provide more insights into the T-cell regulatory role of the TIPBs.

#2 The authors use different dimension reduction techniques (PCA, tSNE). How would other techniques (tSNE, UMAP) perform on the data sets (Fig. 1A, Fig. 3A). Can the different B-cell populations within the TABs also be reproduced by dimension reduction of the flow cytometry data shown in suppl. methods? In essence, this question refers to the robustness of the classification of the different populations of TABs by dimension reduction techniques.

Reviewer #3 (Remarks to the Author): Expert in transcriptomics

The manuscript describes the analysis of various own and published data to estimate the contribution of B cells (in particular, plasmablasts) to the susceptibility and resistance towards checkpoint inhibitor (anti-PD-1) therapy in malignant melanoma. To this end, patient-derived cell culture methods, RNA-Seq, MS-based proteomics and meta-analysis of existing large-scale datasets (e.g. TCGA) are applied.

Overall estimation

Most studies in the past years have been focusing on the interaction between APC and T cells (and the required signaling molecules/pathways) during checkpoint inhibitor therapy. Therefore, an in-depth analysis of the contribution of B cells to resistance is required. In this respect, the manuscript addresses a topical research question, which is important for the understanding of response and resistance to checkpoint inhibitor therapy. The experimental work and data analyses correspond to the state of the art in the field and are described well. In the absence of large amounts of data, the application of meta-analyses is to be acknowledged.

Criticism

Several important conclusions in this manuscript are based on too few (or, in the conditioned medium assay, one) samples only, which raises questions concerning the general validity of the findings. This is supported by the principal components analyses (figs. 1A, 2A) which generally explain less than 50% of the data.

Furthermore, the proteomic and transcriptomic data, which are used to define the TAB and TIPB signatures (which are later applied on large data sets) are supported by less than 5 samples. The definition of the resulting (TAB and TIPB) signatures is not sufficiently explained. Many of the data in this part remain correlative and descriptive. Also, correlation of the TIPB signature with TCGA (fig. S5) and Riaz (fig S6) data does not necessarily imply functional importance but might rather be due to the presence of the certain cell types in the tumor microenvironment.

The TMA analysis shows that only 42% (65/155) of cores (corresponding to how many patients?) are positive. The stringency of this analysis (at least one cell, at least one of the markers) is low.

While intra- and interindividual tumor heterogeneity is large, this finding suggests that a major part of the conclusions remains unexplained.

Recommendation

For the reasons outlined above, this reviewer thinks that the signatures used for defining the B-cell phenotypes might be substantially improved by analyzing a larger number of samples. This addition would lead to an improved stratification of patient responses in the downstream meta-analyses. Functional analyses should be added to show the contribution of the TIPB cells in sustaining the inflammatory state and influencing therapy success.

Reviewers' comments:

We thank the reviewers for the positive response, their instructive comments and clear suggestions, which we think have improved the manuscript significantly. The comments of the reviewers are answered as follows:

Reviewer #1 (Remarks to the Author): Expert in Cancer Immunology

Griss and co-authors use sequencing and immunohistochemistry approaches on new and publically available human datasets to identify a tumor-associated plasmablast-like B cell signature (TIPB) in metastatic melanoma that correlates with patient survival and response to immune checkpoint blockade. Although these data present an interesting potential biomarker for therapy, more mechanistic insight is needed for how tumor associated B cells functionally regulate T cells in the melanoma tumor microenvironment, and how this interaction ultimately may impact clinical response.

1. Is the decrease in TIPB signature on anti-CD20 treatment (Figure 5) associated with decreased overall survival, as may be predicted from data in Figure 4A?

Response: All patients in our clinical study had progressive end-stage melanoma. In lack of a valid comparison group to assess overall survival, we split our patients into two groups based on median survival. As anti-CD20 treatment constantly depleted TAB and subsets from tumor tissues (see Figure 6A, main manuscript) there was no association between the decrease of the TIPB signature and overall survival in these patients.

We therefore repeated this analysis using pre-therapy tumor samples only for a predictive analysis. As a comparison of the molecular TIPB signature had only included 4 samples, we enhanced the number of informative study samples by performing multiplex immunostainings on

all available pre-therapy tumor samples (n=8). Here we compared the frequency of plasmablast-like B cells, CD19+ and CD20+ B cells, as well as CD8+ and CD4+ T cells.

Again, we found no significant difference in cell frequencies for any of the characterised cell types (see figure below).

If so, how might this be explained, given previous studies (cited in Introduction) demonstrating enhanced anti-tumor response and clinical benefit of anti-CD20 treatment in therapy-resistant metastatic melanoma?

Response: Previous observations (including our own) on tumor responses and clinical benefit through B cell-depletion by anti-CD20 antibodies were obtained in end-stage metastatic melanoma patients. There is increasing evidence from autoimmune diseases and other human cancers that the role of B cells (and most likely the immune infiltrate in general) may vary in the different disease stages and contexts (such as therapy resistance), presumably also including the molecular subtype of a cancer.

This phenomenon was shown in murine models of autoimmune diseases such as experimental autoimmune EA, MS and SLE^{1,2}. There, B cell depletion therapy has opposing effects dependent on the stages of the disease. While B cell depletion in the early phase of disease exacerbates disease severity, it attenuates disease severity in the late phase of the disease.

In cancer, most studies did not report sub-analyses based on tumor stage. However, in oro- and hypopharynx cancer CD20+ TIL were reported to be associated with a favourable outcome in early disease but a negative one in advanced disease³. In addition, a favourable prognostic effect of CD20+ TIL was reported to be linked to distinct histologic or molecular subtypes in breast cancer (in ER-, basal, and HER2+, but not triple negative subtype^{4,5}), ovarian cancer (in high-grade serous, but not other subtypes⁶) and mesothelioma (in the epithelioid, but

not non-epithelioid subtype⁷). One possible explanation for these differences may be the relative contribution and timing of B cells with opposing disease-inhibitory and -promoting activities (reviewed in ⁸). So far, however, a concise and generally accepted definition of B cell phenotypes and their associated function at different stages of human melanoma disease progression as well as in response to therapy is missing.

In melanoma, development of resistance to MAPK-inhibitor and ICB therapy are associated with a profound modification of the cellular composition of the tumor microenvironment, including B cells⁹⁻¹². Additionally, genomic alterations profoundly modify the tumor microenvironment including the activation of distinct oncogenic signalling pathways. For example, melanoma-intrinsic WNT/ β -catenin pathway activation precludes recruitment of BAFT3-lineage dendritic cells and thus the cross-presentation of tumor antigens to T cells¹³. In contrast activation of the stimulator of the interferon genes (STING) protein within intratumoral dendritic cells leads to opposing effects^{13,14}.

Together, all these factors may impact the outcome of immunotherapies including anti-CD20 therapy. Here, our functional proteomic and transcriptomic data show that melanoma TAB in general are able to express both pro- and anti-inflammatory factors, important determinants of exerting disease-inhibitory and -promoting activities. Similar to observations in autoimmune diseases, our functional analyses with T cells as well as our data from meta-analyses of pre-therapy melanoma samples argue that melanoma TAB rather promote tumor-associated inflammation before therapy, an important determinant for subsequent response to ICB therapy.

2. Does frequency of intratumoral T cells or T cell subsets (resident, central or effector) change on anti-CD20 therapy? While the reduction in CD8+ T cells in extratumoral stroma is striking in anti-CD20 treatment image (Figure 5D), it is less apparent whether intratumoral T cells are significantly decreased. In addition, cellular density appears to be substantially decreased with anti-CD20 treatment – is this representative of all on-treatment tissues evaluated? Quantification is required.

Response: To address the question about the frequency of extratumoral stromal and intratumoral T cells, we compared pre-therapy with the first on-therapy tumor samples (week 9 \pm 2, 13 samples from 9 patients) by 6 color multiplex immunostaining. Anti-CD20 therapy caused a significant decrease of T-cells at the invasive tumor-stroma margin. Additionally, we

observed a marked reduction of intratumoral T-cells in almost all patients (new Figure 6A, Supplementary Figure 8A). Due to the comparably low number of *intratumoral* T-cells in pre-therapy samples, this change was statistically not significant.

To address the question about the frequency of extratumoral stromal and intratumoral T cell subsets (resident, central or effector), we found 6 pre-therapy samples with a sufficient number of T cells for subset classification. The vast majority of on-therapy samples contained virtually no T cells (with the exception of patient 4, see below). For patient 4, samples over a time frame of nearly two years were available. We therefore performed an additional longitudinal T cell subset analysis in all of this patient's samples - pre-therapy (week 0), on-therapy (weeks 24, 26, 32, where intratumoral T cells were detected) and after-therapy (weeks 84, 87, 98) (new Figure 6B).

T cell subsets were identified by 7 color multiplex immunostaining using the following marker combinations: CD8+CD4-CD69+CD103+ and CD8-CD4+CD69+CD103- cells corresponding to a tissue-resident memory T cell phenotype (T_{RM}); CD8+CD4-CD45RO+CD27+ and CD8-CD4+CD45RO+CD27+ cells corresponding to a central memory T cell phenotype (T_{CM}); CD8+CD4-CD45RO+CD27- and CD8-CD4+CD45RO+CD27- cells corresponding to an effector memory T cell phenotype (T_{EM}); and other CD8+CD4- or CD8-CD4+ T cell phenotypes (Supplementary Figure 9C). The analysis of pre-therapy tumor samples showed a comparable composition of these T cell subtypes between different patients (Supplementary Figure 9A). Additionally, the extended longitudinal time course analysis showed that if T-cells recurred, the overall composition of the T cell subsets was not affected (Supplementary Figure 9B).

We furthermore quantified the total cell counts in all samples pre- and early on-therapy (week 9 ± 2 , 13 samples from 9 patients). There was no change in total cell counts in intra-tumoral tumor areas, but a trend towards a reduction at the invasive tumor-stroma margin which we believe is caused by the marked reduction of immune cells. This change was not statistically significant due to the heterogeneity between individual samples (Supplementary Figure 8B).

Reviewer #2 (Remarks to the Author): Expert in melanoma and immunology

In their study Griss et al. address the role of tumor-associated B-cells (TABs) for the immune contexture of human melanomas. Using a multiplex immunostaining approach with seven established markers the authors characterize TABs in a cohort of human metastatic melanomas. In particular, they describe and focus on a predominant population of CD19+CD20-CD38+CD138-CD27+ plasmablast-like tumour-associated B-cells (TAB). Because TABs were located at the tumor margin, the authors assumed an indirect communication between melanomas cells and TABs. Exposure of autologous B-cells from peripheral blood to melanoma cell-conditioned medium induced a plasmablast-like B-cell phenotype in a NF-kB-dependent manner. The authors therefore suggested that melanoma cells educate B-cells in the tumor microenvironment (TME). Showing that these plasmablast-like TABs produce T-cell recruiting chemokines the authors asked whether TABs, in particular plasmablast-like TABs, are involved in T-cell recruitment.

Analysis of pre- and on-treatment specimens from melanoma patients treated with an anti-CD20 antibody showed reduced infiltration of CD8+ T-cells and macrophages under anti-CD20 therapy.

The work by Griss et al. emphasizes the importance of B-cells for the immune cell composition in melanomas with potential implications for immune checkpoint inhibition as supported by correlative analyses of published gene expression data sets. Indeed, the role of B-cells is largely overlooked in the context of ICI. The authors build on their previous study an expertise in this field. The topic of the present work is timely and of interest to a large scientific community. However, there are several issues that need to be addressed.

-What is the concept of their reduction in on-treatment specimens? Reduced TME-dependent conversion from the general B-cell population?

Response: Plasmablasts are a short-lived cell population which, in tumors, most likely regenerate from tertiary lymphoid structures (TLS) by differentiation through CD20+

precursor-stages of B cells. We therefore analyzed our study samples for the presence of TLS using a 7 color multiplex immunostaining approach for CD20, CD4, CXCL13, CD21, CD23, Bcl6, and DAPI (new Supplementary Figure 8D). In patient-matched tumor samples we observed a significant reduction of the TLS-area upon anti-CD20 therapy (before therapy vs. the first on-therapy tumor samples (week 9±2): $p = 0.04$, new Supplementary Figure 8C). This is currently our best explanation of why the plasmablast-dominated CD20-CD19+ B cell population is reduced through anti-CD20 therapy.

-It is important to provide more patient details related to the study and additional mechanistic insights to TIPBs.

Response: To further elucidate B cell effects on T cell activity, we used a surrogate assay of T cell activity, namely NF- κ B promoter activity in GFP-reporter Jurkat T cells expressing PD-1 (FACS-based analysis, experiment run in triplicates, 4 biological replicates, see Methods, reference). As compared to control (mock)-treated TAB, MCM-treated TAB induced a significant increase of NF- κ B activity in T cells and significantly enhanced NF- κ B activity through PD-1 blockade by pembrolizumab (new Figure 4E). This effect was consistent at two different concentrations of superantigen SEE. As expected, pembrolizumab had no effect in PD-1-negative GFP-reporter Jurkat T cells (not shown).

- How was the reduction of B-cells in the peripheral blood from patients receiving anti-CD20 therapy at the time when biopsies were taken?

Response: Trial patient had been screened for loss of peripheral B cells under treatment. In-line with existing data from the usage of anti-CD20 therapy in rheumatological diseases and lymphoma, peripheral B cells were completely depleted at week 2 (after the second administration of the antibody). At time of the first on-therapy biopsies (week 9±2**), peripheral B cells were no longer detectable in all trial patients. This has been clarified in the manuscript (Methods, patient-derived material, first paragraph).

- What is the time frame between the biopsies?

Response: Pre-therapy samples were acquired immediately before the start of treatment. The first on-therapy sample (which was characterised using RNA-seq) was acquired at week 9±2. This information has been added to the Methods section (patient-derived material, first paragraph). In addition to the RNAseq data we have now performed an additional multiplex immunostaining analysis (CD20, CD19, CD8, CD4, FoxP3, DAPI) of 13 pre- (week 0) and on-therapy (week 9±2) tumor samples from 9 patients for changes in TAB and T cell numbers at the invasive tumor-stroma margin and intratumoral (see new Figure 6A, Supplementary Figure 8A)).

- What about the clinical response? Did patients progress under therapy?

Response: In total, 2 patients showed a stable disease according to ir-RC as best response, while 8 patients showed a progressive disease. These data are available in more detail in doi:10.1038/s41467-017-00452-4.

- Could the reduction of CD8+ T-cells in the tumor microenvironment be related to disease progression rather than a causal relationship to anti-CD20 therapy and B-cell depletion?

Response: We feel that this question can be answered best by data from a longitudinal analysis of 12 tumor samples from one study patient obtained over a time period of 98 weeks. Here we can demonstrate not only a general reduction of B and T cell numbers under anti-CD20 therapy, but also the re-occurrence of intratumoral B and T cells after therapy (new Figure 6B). A short, intermittent increase of extratumoral CD4 and intratumoral CD8 T cells around week 25, respectively, could not be confirmed in later biopsies from the same patient or in tumor samples from two further study patients at comparable time points (new Supplementary Figure 8E). These data argue against the reduction of CD8+ and CD4+ T cells in the tumor microenvironment being related to disease progression.

- What about biological changes of plasmablast-like TABs (TIPBs)? Viability, persistence in the TME? It would be important to characterize biological changes (viability etc...) of B-cells exposed to melanoma-conditioned medium as a surrogate of TIPBs.

Response: As suggested by the reviewer we conducted additional experiments on cell viability of TAB exposed to MCM and compared to control medium. These assays showed increased TAB viability through MCM without affecting cell proliferation as evidenced by total cell numbers over time. The latter observation is in line with our proteomics and RNAseq data from the MCM-induction experiments showing downregulation of cell-cycle associated genes and proteins (new Supplementary Figure 6).

-What about T-cell regulatory properties of TIPBs? Would B-cells exposed to melanoma-conditioned medium exert T-cell suppressive or T-cell stimulating properties in co-culture assays (aCD3/aCD28 stimulations)? This would provide more insights into the T-cell regulatory role of the TIPBs.

Response: We further characterised the mechanistic effect of MCM-conditioned TABs using NF- κ B promoter activity in GFP-reporter Jurkat T cells. This assay showed that MCM-treated TAB significantly increased NF- κ B promoter activity in T cells and significantly enhanced NF- κ B activity through PD-1 blockade by pembrolizumab (please see above, new Figure 4E).

#2 The authors use different dimension reduction techniques (PCA, tSNE). How would other techniques (tSNE, UMAP) perform on the data sets (Fig. 1A, Fig. 3A). Can the different B-cell populations within the TABs also be reproduced by dimension reduction of the flow cytometry data shown in suppl. methods? In essence, this question refers to the robustness of the classification of the different populations of TABs by dimension reduction techniques.

Response: In case of the original Figure 1A we used fixed thresholds and the mentioned antibody signatures (now Figure 1A) to classify B cell subtypes. We merely used the PCA to highlight that this fixed classification also correlates to clusters in an independent PCA. In the revised version of the manuscript we removed the TMA-based analyses (in response to reviewer 3) and replaced it by an analysis of whole tissue sections of 41 melanomas (new Figure 1B).

The classification of subpopulations in the scRNA-seq data was performed using the graph-based clustering algorithm implemented in the R package Seurat. The tSNE-based visualization is completely independent of this approach and therefore provides additional evidence for the classification results.

This was clarified in the methods section of the manuscript (public data sets, third paragraph):

[...] Cells were clustered using Seurath's graph-based clustering algorithm based on the principal components using a resolution of 0.6. [...]

Reviewer #3 (Remarks to the Author): Expert in transcriptomics

The manuscript describes the analysis of various own and published data to estimate the contribution of B cells (in particular, plasmablasts) to the susceptibility and resistance towards checkpoint inhibitor (anti-PD-1) therapy in malignant melanoma. To this end, patient-derived cell culture methods, RNA-Seq, MS-based proteomics and meta-analysis of existing large-scale datasets (e.g. TCGA) are applied.

Overall estimation

Most studies in the past years have been focusing on the interaction between APC and T cells (and the required signaling molecules/pathways) during checkpoint inhibitor therapy. Therefore, an in-depth analysis of the contribution of B cells to resistance is required. In this respect, the manuscript addresses a topical research question, which is important for the understanding of response and resistance to checkpoint inhibitor therapy. The experimental work and data analyses correspond to the state of the art in the field and are described well. In the absence of large amounts of data, the application of meta-analyses is to be acknowledged.

Criticism

-Several important conclusions in this manuscript are based on too few (or, in the conditioned medium assay, one) samples only, which raises questions concerning the general validity of the findings.

Response:

As suggested by the reviewer we have now recapitulated the proteomics data from the induction experiments at the transcript level with MCMs from short term cultured *autologous* melanoma

cells from 4 study patients. This increased the number of tested conditioned media from one to 4 and the number of patients with tested PBMC-B und TAB cells from 4 to 5. Therefore, the proteomics data are now confirmed by 11 individual setups plus individual (mock) controls and a total of 22 RNA-seq-based read-outs. Each of these samples was subjected to a RNA-seq and FACS analysis, which had to be performed separately from proteomics experiments. Thus, the experiments were performed at least two times each (three times in the eight setups where proteomics was performed as well). In summary, the newly generated data is in-line with our previous results (updated Figure 2C). Therefore, we believe that the results shown in the induction experiments are robust.

In addition, we processed all available biopsy samples from our clinical study for the several multiplex immunostaining to quantify different B- and T-cell subpopulations, thus adding 21 tumor samples to the analysis. Therefore, our main observation, the reduction of T-cells through B-cell depletion estimated through RNA-seq data, is now supported through absolute quantification using 7 color multiplex immunostaining in a considerably larger number of clinical samples.

Furthermore, we have now subjected a further whole tumor sections from 41 different patients to a 7 color multiplex immunostaining for CD20, CD19, CD5, CD27, CD38, CD138, and DAPI to further support the signatures used for defining the B-cell phenotypes in the original melanoma TMA.

-This is supported by the principal components analyses (figs. 1A, 2A) which generally explain less than 50% of the data.

Response: The PCA shown in the original Figure 1A was merely used to highlight that the chosen antibody thresholds were suited to separate the different cell types. This was consistent up to component 5 - which we did not show for visual reasons. In the revised version of the manuscript we replaced the TMA-based analysis with the characterisation of whole tissue sections of 41 samples (see below, new Figure 1B).

For Figure 2A we only displayed components 1 and 2 again for visual reasons. Components 3 and 4 also do not separate samples based on cell types. To highlight this fact, we performed an

additional differential expression analysis on the cell types alone, which resulted in no significantly differentially expressed proteins between TAB and PBMCB (all adj. $p > 0.05$). This is now highlighted in the manuscript and the respective analysis code added to the Jupyter notebook:

[...] There was no marked difference between peripheral blood- and tumor-derived B cells with no significantly differentially expressed proteins observed in the proteomics data (Figure 2A).

*[...] (Results, **Human melanoma cells directly induce NF κ B activation in TAB through soluble factors**, second paragraph).*

- Furthermore, the proteomic and transcriptomic data, which are used to define the TAB and TIPB signatures (which are later applied on large data sets) are supported by less than 5 samples.

Response: While the induction experiments were originally performed with B cells from 4 patients, we always used peripheral as well as tumor-derived B cells. Although they are not as independent as different patients, they still must be regarded as distinct biological samples.

As mentioned above we have now analyzed 11 individual setups plus the same number of controls including 5 patients and four different melanoma conditioned media. The newly generated data is in-line with our previous results.

-The definition of the resulting (TAB and TIPB) signatures is not sufficiently explained.

Response: So far, expression of immuno-stimulatory and -suppressive cytokines and cell surface signaling molecules are the most useful factors to distinguish tumor-associated B cell subsets with distinct functions. We therefore first identified cytokines and cell surface signaling molecules with significant induction on protein and transcript levels in the MCM-induction experiments. Thereafter, induced proteins/genes were classified for immune-associated activities according to published functional data in lymphocytes, whenever available in B cells, and those with related activity grouped into cohorts. Following this procedure we set up 6 different cohorts characterized by expression of proteins/genes associated with key activities critically determining tumor inflammation and the outcome of anti-tumor immune responses^{15,16}, namely immune activation, co-stimulation, inflammation, immuno-suppression, B cell exhaustion and immune checkpoint regulation.

We explicitly chose this very conservative and certainly manual approach as we felt that it is vital that all of our signatures have a literature-based background. Recently, several signatures were published that are purely “data-driven”. These “data-driven” signatures are often very large and hard to completely understand from a biological perspective.

Our signatures have the additional advantage that they can be more easily transferred into a clinical setting where immunohistochemistry is an established routine procedure. Even though our approach is manual and thus potentially biased by our own knowledge of tumor immunology, we feel that we were rigorous in the validation of these signatures using a multitude of independent datasets. Therefore, we feel that the advantage of having concise, well characterised signatures to define our cell populations of interest outweighs the disadvantage of the chosen manual approach.

-Many of the data in this part remain correlative and descriptive. Also, correlation of the TIPB signature with TCGA (fig. S5) and Riaz (fig S6) data does not necessarily imply functional importance but might rather be due to the presence of the certain cell types in the tumor microenvironment.

Response: To increase the functional evidence we performed additional *in-vitro* experiments as also suggested by reviewer#2. As suggested we performed several additional functional experiments to elucidate (i) the effects of MCM on plasmablast-like TAB viability and proliferation as well as (ii) T-cell suppressive or T-cell stimulating properties of MCM-induced plasmablast-like TAB in co-culture assays with T cells. These assays showed an increased viability through MCM (new Supplementary Figure 6). Furthermore, MCM-treated TAB induced a significant increase of NF- κ B activity in T cells as compared to control (mock)-treated TAB, and significantly enhanced NF- κ B activity through PD-1 blockade by pembrolizumab (new Figure 4E).

Even though our analysis of the TCGA and Riaz et al. dataset do not prove a direct functional importance, they do support our proposed mechanism. The newly created functional *in-vitro* data furthermore supports the observed correlations.

Finally, the single-cell data by Sade-Feldman *et al.* clearly shows that all of our proposed functionally characterised molecules are indeed expressed by TAB in human melanoma *in vivo*.

-The TMA analysis shows that only 42% (65/155) of cores (corresponding to how many patients?) are positive. The stringency of this analysis (at least one cell, at least one of the markers) is low. While intra- and interindividual tumor heterogeneity is large, this finding suggests that a major part of the conclusions remains unexplained.

Response: The heterogeneous distribution of immune cells not only within a single tumor but also across different tumors within a single and between different patients is well documented. This naturally complicates the quantification of immune cell infiltration, particularly in TMA-based analyses. This is amplified by the fact that our TMA cores were not explicitly created to contain preferentially stromal tumor areas where TAB predominantly reside.

To reduce the TMA-based bias we now analysed whole tissue sections from 41 human melanomas (new Figure 1B). While there still is considerable variation in the exact composition of TAB subtypes between the different samples, the key subtypes described by us are present in >90% of the samples.

Recommendation

For the reasons outlined above, this reviewer thinks that the signatures used for defining the B-cell phenotypes might be substantially improved by analyzing a larger number of samples.

This addition would lead to an improved stratification of patient responses in the downstream meta-analyses.

Response: As suggested, we considerably increased the number of samples in the induction experiments which supported our existing functional B cell signatures (see above). In addition, we extended our analysis to all available study samples from the clinical trial (21 tumor samples, see also new Figure 6) and added an analysis of whole tumor sections from additional 41 samples to support the identified B cell phenotypes. This has the additional advantage that cell type abundance changes through anti-CD20 therapy is now not only estimated based on whole-tissue RNA-seq data but confirmed through absolute, immunohistochemistry based cell quantification.

Functional analyses should be added to show the contribution of the TIPB cells in sustaining the inflammatory state and influencing therapy success.

Response: As already mentioned we performed additional functional analyses to characterise the effect of MCM on the viability of B cells and of MCM-conditioned B cells on T cell function (see new Supplementary Figure 6, new Figure 4E).

References

1. Matsushita, T., Yanaba, K., Bouaziz, J.-D., Fujimoto, M. & Tedder, T. F. Regulatory B cells inhibit EAE initiation in mice while other B cells promote disease progression. *J. Clin. Invest.* **118**, 3420–3430 (2008).
2. Haas, K. M. *et al.* Protective and pathogenic roles for B cells during systemic autoimmunity in NZB/W F1 mice. *J. Immunol.* **184**, 4789–4800 (2010).
3. Distel, L. V. *et al.* Tumour infiltrating lymphocytes in squamous cell carcinoma of the oro- and hypopharynx: prognostic impact may depend on type of treatment and stage of disease. *Oral Oncol.* **45**, e167–74 (2009).
4. Mahmoud, S. M. A. *et al.* The prognostic significance of B lymphocytes in invasive carcinoma of the breast. *Breast Cancer Res. Treat.* **132**, 545–553 (2012).
5. Song, I. H. *et al.* Predictive Value of Tertiary Lymphoid Structures Assessed by High Endothelial Venule Counts in the Neoadjuvant Setting of Triple-Negative Breast Cancer. *Cancer Res. Treat.* **49**, 399–407 (2017).
6. Milne, K. *et al.* Systematic analysis of immune infiltrates in high-grade serous ovarian cancer reveals CD20, FoxP3 and TIA-1 as positive prognostic factors. *PLoS One* **4**, e6412 (2009).
7. Chee, S. J. *et al.* Evaluating the effect of immune cells on the outcome of patients with mesothelioma. *Br. J. Cancer* **117**, 1341–1348 (2017).
8. Matsushita, T. Regulatory and effector B cells: Friends or foes? *J. Dermatol. Sci.* **93**, 2–7 (2019).
9. Hugo, W. *et al.* Non-genomic and Immune Evolution of Melanoma Acquiring MAPKi Resistance. *Cell* **162**, 1271–1285 (2015).
10. Hugo, W. *et al.* Genomic and Transcriptomic Features of Response to Anti-PD-1 Therapy in Metastatic Melanoma. *Cell* **165**, 35–44 (2016).
11. Taube, J. M. *et al.* Association of PD-1, PD-1 ligands, and other features of the tumor immune microenvironment with response to anti-PD-1 therapy. *Clin. Cancer Res.* **20**, 5064–5074 (2014).
12. Koyama, S. *et al.* Adaptive resistance to therapeutic PD-1 blockade is associated with upregulation of alternative immune checkpoints. *Nat. Commun.* **7**, 10501 (2016).

13. Spranger, S., Bao, R. & Gajewski, T. F. Melanoma-intrinsic β -catenin signalling prevents anti-tumour immunity. *Nature* **523**, 231–235 (2015).
14. Corrales, L., McWhirter, S. M., Dubensky, T. W., Jr & Gajewski, T. F. The host STING pathway at the interface of cancer and immunity. *J. Clin. Invest.* **126**, 2404–2411 (2016).
15. Chen, D. S. & Mellman, I. Elements of cancer immunity and the cancer-immune set point. *Nature* **541**, 321–330 (2017).
16. Fridman, W. H., Zitvogel, L., Sautès-Fridman, C. & Kroemer, G. The immune contexture in cancer prognosis and treatment. *Nat. Rev. Clin. Oncol.* **14**, 717–734 (2017).

REVIEWERS' COMMENTS:

Reviewer #1 (Remarks to the Author):

The authors have thoroughly addressed reviewer comments, in part through addition of more patient samples to their analyses.

In addition, the authors' additional 'Discussion' text related to different functional roles for B cells in different phases of disease progression somewhat helps to explain how B cells may promote anti-tumor immunity in metastatic melanoma (as they observe in this manuscript), whereas B cell depletion with aCD20 therapy could be clinically beneficial in other contexts (as they have previously published in a cohort of end-stage metastatic melanoma patients).

Reviewer #2 (Remarks to the Author):

The authors addressed all my comments and the manuscript is now suitable for publication.

Reviewer #3 (Remarks to the Author):

My previous concerns have been adequately addressed by the authors.